# You Only Prune Once: A Zero-Shot, Data-Free Pruning at Initialization with Transferable Supermasks

## Abstract

Pruning at Initialization (PaI) accelerates training while maintaining accuracy, yet most criteria depend on data and backpropagation, leaving them brittle. Slight variations in random seed or sparsity budget reorder scores require rescoring or iterative schedules and yield masks with weak transferability across seeds, datasets, and budgets. The proposed *You Only Prune Once* (**YOPO**) framework addresses these limitations through a *zero-shot*, data and gradient-free design. YOPO computes a *once-only* saliency by fitting a nonnegative low-rank model to absolute weights at random initialization and measuring the element-wise Frobenius residual. Global or layer-wise thresholds generate masks with *exact* sparsity control and no layer collapse. Since ordering and budget are decoupled, the same saliency supports *re-thresholding* to any sparsity and *dataset transfer* without rescoring, enabling reusable "supermasks". Experiments on CIFAR-10, CIFAR-100, Tiny-ImageNet, and ImageNet with standard CNN backbones show that YOPO matches or surpasses strong single-shot PaI baselines, rivals iterative/data-dependent methods despite using no data at initialization, and consistently outperforms expander-graph zero-shot PaI. Altogether, YOPO provides a scalable and intuitive approach to initialization-time pruning with stable transfer across seeds, datasets, and sparsity levels.

## 1 Introduction

Modern neural networks are often heavily overparameterized, which invites structured or unstructured pruning to reduce memory, computation, and training time without sacrificing accuracy. An especially attractive paradigm is PaI: remove parameters *before* learning begins and train the sparse subnetwork from scratch under a standard recipe. The conceptual root is the Lottery Ticket Hypothesis (LTH), which posits that randomly initialized dense models contain sparse "winning tickets" that can be trained in isolation from their *initial* weights to match dense performance (Frankle & Carbin, 2019). Early demonstrations relied on iterative magnitude pruning with weight rewinding, effective but computationally heavy. This motivated single-shot criteria that score connections directly at initialization. Among these, SNIP evaluates the loss sensitivity of each weight using data and first-order gradients (Lee et al., 2019), while GraSP incorporates Hessian information to preserve gradient flow (Wang et al., 2020). A complementary line, SynFlow, enforces a positivity-based conservation law to avoid layer collapse in a *data-free* fashion, but does so through an iterative scoring schedule (Tanaka et al., 2020).

Although these approaches advance the PaI paradigm, two practical frictions remain. *First*, most criteria are inherently *data and seed dependent*: saliencies are computed from task-specific batches and are sensitive to the random initialization, so masks often need to be recomputed when the dataset changes, when the target sparsity (budget) changes, or even when the seed changes. *Second*, recent sanity-check studies argue that a considerable portion of PaI gains can arise from *how sparsity is allocated across layers* rather than from precise per-weight ranking (Su et al., 2020; Ma et al., 2021). This raises a sharper question: can we construct an *initialization-time* saliency that is both *data-free* and *once-only*, thereby enabling *reuse* across datasets and across sparsity budgets by mere re-thresholding, while remaining competitive with data-dependent baselines?

Two recent directions illuminate the landscape. One replaces saliency with *connectivity priors*: expander-style $d$-regular bipartite masks instantiated at initialization deliver strong signal propagation and competitive accuracy at extreme sparsities, consistently outperforming random wiring and earlier expander variants (Stewart et al., 2023). These approaches are appealingly data-free, yet they do not by themselves address transfer across datasets or across budgets and introduce graph-construction choices (e.g., degree $d$, generator family) that can complicate deployment. At the other end of the spectrum, *score optimization at initialization* ("GEM-MINER") learns supermasks that rival prune-after-train baselines and pass modern sanity checks (Sreenivasan et al., 2022), but requires labeled data and many epochs of score training. A data-agnostic perspective argues for topology-guided criteria less entangled with dataset statistics Pham et al. (2023); concurrently, theory delineates when purely data-free PaI can (or cannot) succeed Kumar et al. (2024), and recent results sharpen sparse variants of the strong Lottery Ticket Hypothesis Natale et al. (2024). Surveys synthesize these developments and chart open problems in PaI, from sanity-check robustness to deployability and efficiency trade-offs Cheng et al. (2024); Wang et al. (2022).

Positioned between these extremes, the proposed method **YOPO** follows a once-only, data and gradient-free route that explicitly targets *transferability*. A single, dataset-agnostic saliency per backbone is obtained by fitting a *nonnegative low-rank template* to the absolute weights and scoring each connection via its *elementwise Frobenius residual*. For a prunable layer with absolute weights $\tilde{W} = |W|$, the factorization $\tilde{W} \approx VH$ with $V, H! \geq !0$ is computed using Nonnegative Matrix Factorization (NMF), and saliency is defined as $S = |\tilde{W} - VH|$. The additive, parts-based reconstruction $VH$ captures low-rank regularities, whereas large residuals highlight distinctive, potentially informative connections (Lee & Seung, 1999; 2001). A robust monotone re-thresholding scheme is applied layer-wise or globally, convert $S$ into masks that realize an exact sparsity budget in a *single shot* while avoiding collapses. Since the ordering in $S$ is fixed, any sparsity level emerges through re-thresholding, and the resulting mask family transfers seamlessly across datasets without rescoring.

**The main contributions of this research are summarized as follows:**

(i) A *zero-shot*, data and gradient-free PaI criterion based on *nonnegative low-rank residuals*, providing a simple alternative to data-dependent initialization pruning.

(ii) Formalization and systematic evaluation of *mask transferability*, showing that a once-only ordering enables reusable "super tickets" across datasets and sparsity levels.

(iii) A lightweight *collapse-avoidance* principle: enforcing nonnegativity with robust thresholds ensures per-neuron survival in a single shot without iterative scoring.

(iv) Comprehensive experiments on CIFAR-10/100, Tiny-ImageNet, and ImageNet with VGG, ResNet, and WRN backbones, demonstrating consistent outperformance of existing zero-shot PaI methods, competitive results against data-dependent baselines, and unique support for budget reuse and dataset transfer without restoring.

Therefore, the proposed YOPO reframes PaI through the lens of *once-only*, data-free saliency derived from nonnegative low-rank structure: it simplifies initialization-time computation, provides exact budget control, and most importantly decouples mask selection from downstream datasets, enabling practical *reuse* across tasks and sparsity levels.

## 2 PROBLEM FORMULATION AND BACKGROUND

Let $\mathcal{A}$ be a fixed architecture with parameter vector $\theta \in \mathbb{R}^n$ and random initialization $\theta_0 \sim \mathcal{P}_{\text{init}}$. We consider a family of supervised datasets $\mathcal{D} = \{D_1, \ldots, D_T\}$ drawn from a task distribution $\mathsf{T}$. A *binary mask* $m \in \{0, 1\}^n$ induces a pruned model $\theta_0 \odot m$ (Hadamard product). Training from scratch on $D \in \mathcal{D}$ with a fixed recipe produces parameters $\theta^\star(m; D)$ and validation risk $\mathcal{R}(m; D)$ (or accuracy $\text{Acc}(m; D)$). Global sparsity is

$$s(m) = 1 - \frac{\|m\|_0}{n} \in [0, 1].$$

The mask $m$ is *trainable on $D$ at sparsity $s$* if $\text{Acc}(m; D)$ is within a small tolerance $\varepsilon$ of the dense baseline under the same training protocol.

**Classical PaI as a data-dependent mapping.** PaI can be abstracted as a scoring rule followed by a budgeted selection. Given $(\theta_0, D, p)$, many PaI methods compute a data-dependent saliency $\sigma(\theta_0; D) \in \mathbb{R}^n$ and return a mask

$$f(\mathcal{A}, D, p) \;=\; m_{D,p} \;:=\; \mathrm{TopK}\big(\sigma(\theta_0; D),\, k(p)\big), \qquad k(p) = (1-p)\, n, \tag{1}$$

where TopK keeps the $k(p)$ largest entries (ties broken arbitrarily). SNIP and GraSP instantiate $\sigma$ with gradient- and Hessian-based criteria computed on batches from $D$ (Lee et al., 2019; Wang et al., 2020); SynFlow constructs $\sigma$ through an iterative, data-agnostic flow conservation process (Tanaka et al., 2020). Two consequences follow from equation 1: *(i)* for each *new dataset $D'$* the full scoring procedure must be repeated to obtain $m_{D',p}$; *(ii)* for each *new sparsity level $p'$* one recomputes or at least re-ranks saliency to target the new budget and obtain $m_{D,p'}$.

**Once-only, dataset-agnostic saliency.** We aim to decouple mask construction from data and sparsity by computing a *single* saliency vector

$$\underbrace{S(\theta_0) \in \mathbb{R}_{\geq 0}^n}_{\text{computed once, no data/gradients}} \tag{2}$$

and deriving masks for *any* target sparsity by re-thresholding/ranking *without recomputing scores*. Formally, define a family of *threshold operators* $\{\mathcal{T}_\tau\}_{\tau \geq 0}$ acting elementwise:

$$\mathcal{T}_\tau[S]_i \;=\; \mathbb{K}\{S_i > \tau\}, \qquad i \in \{1, \ldots, n\}. \tag{3}$$

A *once-only* scheme selects $\tau = \tau(p)$ so that $s(\mathcal{T}_{\tau(p)}[S]) = p$ (up to a small tolerance), yielding

$$m_p \;=\; \mathcal{T}_{\tau(p)}[S(\theta_0)] \;\in\; \{0,1\}^n. \tag{4}$$

Equivalently, one can view $m_p = \mathrm{TopK}(S(\theta_0), k(p))$. The crucial property is that *all* masks across $p \in [0,1]$ are induced by a *fixed* ranking of $S(\theta_0)$.

**Transferability objective.** Let $\mathcal{P} \subset [0,1]$ be a set of sparsity budgets of interest and $\mathcal{D}_{\mathrm{tgt}} \subseteq \mathcal{D}$ a set of target datasets. The *supermask* objective seeks an $S(\theta_0)$ such that the induced masks $m_p$ are simultaneously trainable across datasets and budgets:

$$\min_{S \in \mathbb{R}_{\geq 0}^n} \;\; \mathbb{E}_{D \sim \mathsf{T}} \Big[ \sup_{p \in \mathcal{P}} \Delta(m_p; D) \Big] \quad \text{where} \quad \Delta(m; D) = \mathcal{R}(m; D) - \mathcal{R}(\mathbf{1}; D), \tag{5}$$

subject to the *once-only* constraint equation 4. An $\varepsilon$-*supermask* satisfies $\forall D \in \mathcal{D}_{\mathrm{tgt}},\ \forall p \in \mathcal{P} :$ $\Delta(m_p; D) \leq \varepsilon$, with training performed from scratch on $(\theta_0 \odot m_p, D)$ under a fixed recipe. This formulation captures two desiderata: *dataset transfer* ($D$ varies) and *budget reuse* ($p$ varies) *without* recomputing scores or using extra budget (epochs).

**Background: why once-only helps.** equation 4 separates the *ordering* of parameters (encoded in $S$) from the *budget* $p$ (encoded in $\tau$). In contrast, data-dependent PaI implements $f(\mathcal{A}, D, p)$ as in equation 1, entangling $D$ and $p$ with $\sigma$. When $S$ is dataset-agnostic and computed once, changing $p$ reduces to threshold selection (no recomputation), and moving from $D_i$ to $D_j$ reuses the same $m_p$ or simply re-thresholds $S$ to a new budget. This matches the operational goal stated informally as

$$\underbrace{f(\mathcal{A})}_{\text{once}} \;\Rightarrow\; \underbrace{\{m_p\}_{p \in \mathcal{P}}}_{\text{all budgets}} \quad \text{transferable across} \quad \mathcal{D}_{\mathrm{tgt}},$$

in contrast to $f(\mathcal{A}, D, p)$ which must be re-evaluated for each $(D, p)$ pair.

## 3 PROPOSED METHOD: YOPO (YOU ONLY PRUNE ONCE)

YOPO is a *zero-shot, data, and gradient-free* PaI method. Given a randomly initialized network, it computes a *once-only*, dataset-agnostic saliency from a nonnegative low-rank reconstruction of absolute weights and produces a binary mask by robust thresholding to match a target sparsity. The same *once-only* saliency supports *multiple sparsity budgets* and *transfers across datasets* without recomputation.

## 3.1 NOTATION AND PRELIMINARIES

Let $\mathcal{A}$ be a fixed architecture with prunable layers $\mathcal{L}$. For a convolutional layer $\ell$,

$$W^{(\ell)} \in \mathbb{R}^{o_\ell \times i_\ell \times k_\ell^h \times k_\ell^w}, \qquad d_\ell = i_\ell k_\ell^h k_\ell^w, \tag{6}$$

and define the flatten/unflatten operators

$$\text{flat}: \mathbb{R}^{o \times i \times k^h \times k^w} \to \mathbb{R}^{o \times d}, \quad \text{unflat}: \mathbb{R}^{o \times d} \to \mathbb{R}^{o \times i \times k^h \times k^w}.$$

We work with elementwise absolute weights $\tilde{W}^{(\ell)} \triangleq \left| \text{flat}(W^{(\ell)}) \right| \in \mathbb{R}_{\geq 0}^{o_\ell \times d_\ell}$. A binary mask $M^{(\ell)} \in \{0,1\}^{o_\ell \times d_\ell}$ prunes via $W^{(\ell)} \leftarrow W^{(\ell)} \odot \text{unflat}(M^{(\ell)})$ and we freeze pruned parameters during training with $\nabla W^{(\ell)} \leftarrow \nabla W^{(\ell)} \odot \text{unflat}(M^{(\ell)})$. Global sparsity is $s = 1 - \frac{\sum_\ell \|M^{(\ell)}\|_0}{\sum_\ell o_\ell d_\ell}$.

## 3.2 ONCE-ONLY SALIENCY VIA NONNEGATIVE LOW-RANK RESIDUALS

We construct a *dataset-agnostic* saliency *once* at initialization by measuring how poorly the absolute weights of each layer are explained by a small additive (nonnegative, low-rank) Frobenius template. Let $\ell \in \mathcal{L}$ index a prunable layer with flattened, nonnegative weights

$$\tilde{W}^{(\ell)} \triangleq \left| \text{flat}(W^{(\ell)}) \right| \in \mathbb{R}_{\geq 0}^{o_\ell \times d_\ell}.$$

We approximate $\tilde{W}^{(\ell)}$ by a nonnegative rank–$r_\ell$ model and define the *elementwise residual* as saliency.

**Nonnegative low-rank model.** We solve the Euclidean NMF (Lee & Seung, 1999; 2001) problem

$$\min_{V^{(\ell)} \geq 0, \ H^{(\ell)} \geq 0} \ \left\| \tilde{W}^{(\ell)} - V^{(\ell)} H^{(\ell)} \right\|_F^2, \qquad V^{(\ell)} \in \mathbb{R}_{\geq 0}^{o_\ell \times r_\ell}, \ H^{(\ell)} \in \mathbb{R}_{\geq 0}^{r_\ell \times d_\ell}. \tag{7}$$

We employ the standard multiplicative updates (Lee & Seung, 1999; 2001) with a small $\varepsilon > 0$:

$$V \leftarrow V \odot \frac{\tilde{W} H^\top}{V H H^\top + \varepsilon}, \qquad H \leftarrow H \odot \frac{V^\top \tilde{W}}{V^\top V H + \varepsilon}, \tag{8}$$

initialized either by nonnegative SVD seeding or i.i.d. $\text{Uniform}[0, 1]$.

**Elementwise residual saliency.** Given any feasible factors $(V^{(\ell)}, H^{(\ell)})$ (after $T$ iterations of equation 8), define

$$S^{(\ell)} \triangleq \left| \tilde{W}^{(\ell)} - V^{(\ell)} H^{(\ell)} \right| \in \mathbb{R}_{\geq 0}^{o_\ell \times d_\ell}, \tag{9}$$

and unflatten to the native tensor shape for masking. Intuitively, the nonnegative product $VH$ captures *additive, parts-based* regularities; the residual highlights idiosyncratic, less-template-like connections that we retain (Lee & Seung, 1999). Variants include replacing the Euclidean loss in equation 7[1] with KL or Itakura–Saito divergences, and using a per-output-channel factorization that reduces each convolution row to $1 \times d_\ell$ to accelerate preprocessing while preserving rankings.

STRUCTURAL PROPERTIES

We record simple properties that explain the "once-only" and robustness behavior.

**Proposition 1** (Magnitude and perfect-template limits). *Let $S^{(\ell)}$ be defined by equation 9.*

 *(a) (Rank-0 reduction) If $r_\ell = 0$ (i.e., $VH \equiv 0$), then $S^{(\ell)} = \tilde{W}^{(\ell)}$; YOPO reduces to magnitude ranking at initialization.*

 *(b) (Exact template) If $\text{rank}_+(\tilde{W}^{(\ell)}) \leq r_\ell$ and $VH = \tilde{W}^{(\ell)}$ exactly, then $S^{(\ell)} \equiv 0$. In practice this is a measure-zero event; we prevent degenerate collapse by (i) choosing small $r_\ell$, (ii) using strict thresholds, and (iii) enforcing per-row minimum keep.*

---

[1]We also tried KL and Itakura–Saito updates; results were similar while Euclidean was faster to compute.

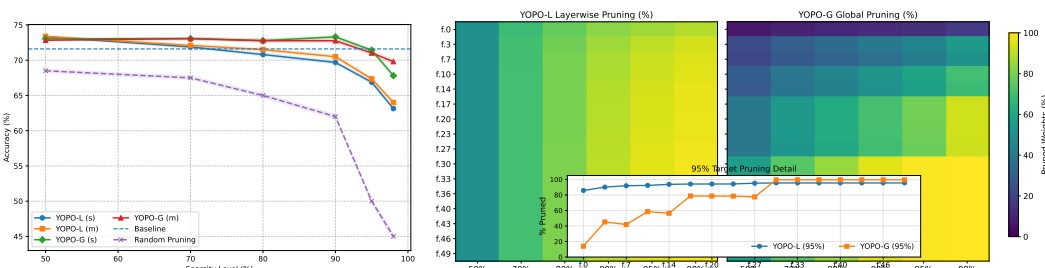

**(a)** YOPO variants vs. random pruning across sparsity.

**(b)** Layerwise vs. global sparsity allocation on VGG19.

**Figure 1: YOPO performance and sparsity allocation profiles.** (a) Across all sparsity levels, YOPO variants outperform random pruning with lower variance. (b) Heatmaps show that YOPO-L enforces near-uniform pruning, whereas YOPO-G adapts sparsity across depth—preserving early layers (14–45% pruned) and aggressively pruning deeper layers ($> 95\%$).

**Lemma 1** (Positive homogeneity and scale stability). *For any scalar $c > 0$, replacing $\tilde{W}^{(\ell)}$ by $c\tilde{W}^{(\ell)}$ admits optimal factors $(\sqrt{c}V^{\star}, \sqrt{c}H^{\star})$ whenever $(V^{\star}, H^{\star})$ is optimal for $\tilde{W}^{(\ell)}$, and the corresponding residual scales as $S^{(\ell)} \mapsto c\,S^{(\ell)}$. Hence the* ranking *of entries in $S^{(\ell)}$ is invariant to positive scalar rescaling of a layer.*

**Lemma 2** (Permutation invariance within flattened structure). *Let $\Pi_r, \Pi_c$ be permutation matrices acting on rows/columns of $\tilde{W}^{(\ell)}$ (e.g., re-indexing output channels or kernel coordinates under flattening). Then $S^{(\ell)}$ permutes accordingly: if $\tilde{W}' = \Pi_r\tilde{W}^{(\ell)}\Pi_c$, an NMF of $\tilde{W}'$ yields $S' = \Pi_r S^{(\ell)}\Pi_c$. Thus once-only rankings respect reindexings inherent to convolutional reshaping.*

**Proposition 2** (Nested masks across budgets). *Let $M^{(\ell)}(\tau) = \mathbf{1}[S^{(\ell)} > \tau]$. If $\tau_1 < \tau_2$, then $M^{(\ell)}(\tau_2) \leq M^{(\ell)}(\tau_1)$ elementwise. Consequently, the sets of survivors are nested as sparsity increases, and all budgets $p \in [0, 1]$ are obtained by* re-thresholding *a fixed $S^{(\ell)}$ without recomputation.*

### WHY NONNEGATIVITY? COMPARISON TO SVD/CP/TUCKER

Low-rank SVD approximations can cancel positive and negative components, obscuring which *entries* are systematically explained by a small additive template. In contrast, NMF is *parts-based*: reconstructions are additive and nonnegative, and any mismatch must appear as a positive residual (used in our collapse argument). CP/Tucker compress dense tensors but do not enforce nonnegativity; their residuals are not guaranteed to be elementwise informative in the same way. Empirically, we found that NMF residuals yield stable rankings at initialization and high kernel sparsity at large global $s$.

### 3.3 ROBUST THRESHOLDING AND CALIBRATION

Given the once-only saliency $S^{(\ell)}$ (Sec. 3.2), YOPO converts scores into a binary mask through a *robust* threshold per layer. We use either a median/MAD cutoff or a mean/STD cutoff:

$$\text{MAD:} \qquad \tau_\ell(\alpha) = \text{median}\big(S^{(\ell)}\big) + \alpha \cdot \text{MAD}\big(S^{(\ell)}\big), \tag{10}$$

$$\text{STD:} \qquad \tau_\ell(\alpha) = \mu\big(S^{(\ell)}\big) + \alpha \cdot \sigma\big(S^{(\ell)}\big), \tag{11}$$

and define

$$M^{(\ell)}(\alpha) = \mathbf{1}\big[S^{(\ell)} > \tau_\ell(\alpha)\big] \in \{0, 1\}^{o_\ell \times d_\ell}, \qquad \hat{s}(\alpha) = 1 - \frac{\sum_\ell \|M^{(\ell)}(\alpha)\|_0}{\sum_\ell o_\ell d_\ell}. \tag{12}$$

The parameter $\alpha \in \mathbb{R}_{\geq 0}$ tunes aggressiveness. We adopt strict comparison ($>$) to avoid tie ambiguity; in practice, ties are rare and can be broken deterministically by infinitesimal jitter. Furthermore, YOPO supports two calibration regimes:

- **YOPO-G (global).** A *single* $\alpha$ is chosen so that $\hat{s}(\alpha) \approx s$ for a given target sparsity $s$, via 1-D bisection. This uses the same $\alpha$ for all layers.

- **YOPO-L (local).** Each layer receives its *own* $\alpha_\ell$ to meet a per-layer budget $s_\ell$ with $\sum_\ell s_\ell o_\ell d_\ell = s \sum_\ell o_\ell d_\ell$. A simple and effective choice is the *equal-ratio* policy $s_\ell \equiv s$ (i.e., maintain approximately the same sparsity ratio in every layer).

Both calibration regimes incorporate the *survival constraints* to prevent row/column collapse.

$$\sum_{j=1}^{d_\ell} M_{ij}^{(\ell)} \geq m_{\min}, \qquad \sum_{i=1}^{o_\ell} M_{ij}^{(\ell)} \geq c_{\min} \quad \text{(typically } m_{\min} \in \{1, 2\}, \; c_{\min} \in \{0, 1\}), \qquad (13)$$

We establish sufficient conditions under which YOPO guarantees per-neuron (row) and per-layer survival, and provide full proofs in Appendix A (see Prop. 3 and Cor. 1).

**Monotonicity and Exact Budget Control:** For both thresholding rules in equation 10 & equation 11, the achieved sparsity $\hat{s}(\alpha)$ is *monotone non-decreasing* in $\alpha$ because masks are nested: if $\alpha_2 > \alpha_1$ then $M^{(\ell)}(\alpha_2) \leq M^{(\ell)}(\alpha_1)$ elementwise (Prop. 2). Consequently, a 1-D bisection on $\alpha$ attains any feasible budget to tolerance $\varepsilon$ in $\mathcal{O}(\log(1/\varepsilon))$ steps. We instantiate this as *MORT* (Monotone Re-Thresholding): (i) for **YOPO-G**, we pool scores across layers and select a *single* $\alpha$ to match the global sparsity; (ii) for **YOPO-L**, we solve independent per-layer bisections (with the *min-keep* row constraint to preclude collapse). See Alg. 2 for the procedure.

**Robustness and calibration.** At high sparsity, $S^{(\ell)}$ is often heavy–tailed; MAD thresholds are therefore more stable than STD in practice (cf. Figs. 1a, 4). One may equivalently standardize via $Z^{(\ell)} = (S^{(\ell)} - \text{median})/(\text{MAD} + \delta)$ and threshold $Z^{(\ell)} > \alpha$, though we adopt the direct MAD form in equation 10 for simplicity. Among calibrations, **YOPO-G** typically attains slightly higher accuracy than **YOPO-L** at matched global sparsity by allocating more parameters to layers with larger residual statistics (Fig. 1a); **YOPO-L** enforces near-uniform per-layer ratios, while **YOPO-G** prunes gently in early layers and more aggressively in deeper ones (heatmap in Fig. 1b), aligning with the heuristic of preserving low-level features. Additional analyses appear in Sec. 4, with complexity and practical guidance in Appx. G.

### 3.4 TRANSFERABILITY AND BUDGET REUSE FROM A ONCE-ONLY SALIENCY

In Section 2, the transferability objective is introduced. Building on this, our goal is to separate the process of mask construction from the specifics of the data and the sparsity level. To achieve this, we rely on a *once-only* saliency equation 4 and re-thresholding equation 12, which serves as a reusable foundation for the rest of our approach.

**Once-only family of masks.** Let $S(\theta_0) = \{S^{(\ell)}\}_{\ell \in \mathcal{L}}$ be YOPO's layerwise saliency computed *once* at initialization (Sec. 3.2). For any sparsity $p \in [0, 1]$, define the global mask family

$$\mathcal{M}(p) = \left\{ M^{(\ell)}(\tau_\ell(\alpha(p))) \; \middle| \; \ell \in \mathcal{L} \right\}, \quad \text{where} \quad \tau_\ell(\alpha) = \text{median}(S^{(\ell)}) + \alpha \, \text{MAD}(S^{(\ell)}). \quad (14)$$

By monotonicity, there exists $\alpha(p)$ with achieved sparsity $\hat{s}(\alpha(p)) \approx p$; the *ordering* of entries within each $S^{(\ell)}$ is fixed across all $p$. The local variant $\mathcal{M}_L(p)$ uses per-layer $\alpha_\ell(p)$ to meet per-layer quotas. In both regimes, no saliency recomputation is needed when $p$ changes: $m_p = \text{TopK}(S, k(p))$ or equivalently $m_p = \mathbf{1}[S > \tau(p)]$.

**Dataset transfer.** Because $S(\theta_0)$ depends only on $|\theta_0|$ and a low-rank additive template, the same family $\{\mathcal{M}(p)\}_{p \in [0,1]}$ is applicable to any $D$ drawn after initialization. Let training from scratch on $(\theta_0 \odot \mathcal{M}(p), D)$ yield validation risk $\mathcal{R}(p; D)$ and dense risk $\mathcal{R}_{\text{dense}}(D)$. We evaluate transfer via two indices.

**Definition 1** (Dataset-Transfer Index (DTI)). *Fix a source dataset $D_{\text{src}}$ and let $\mathcal{M}_{\text{src}}(p)$ denote the mask family constructed from $S(\theta_0)$ (and, if desired, a concrete mask at a particular $p$). For any target $D_{\text{tgt}}$ and budget $p$,*

$$\text{DTI}(D_{\text{src}} \to D_{\text{tgt}}, p) = \mathcal{R}\big(p; D_{\text{tgt}} \mid \mathcal{M}_{\text{src}}(p)\big) - \mathcal{R}\big(p; D_{\text{tgt}} \mid \mathcal{M}_{\text{tgt}}(p)\big),$$

---

**Algorithm 1** YOPO: You Only Prune Once (Pseudocode)

---

**Require:** Random init $\{W^{(\ell)}\}_{\ell \in \mathcal{L}}$; target sparsity $s$ (global) or budgets $\{s_\ell\}$ (local); ranks $\{r_\ell\}$; min-keep $(m_{\min}, c_{\min})$

---

1: **for** $\ell \in \mathcal{L}$ **do**                                                 ▷ For each layer compute the saliency
2:      $\tilde{W}^{(\ell)} \leftarrow \left| \text{flat}(W^{(\ell)}) \right|$                                 ▷ optional: layerwise normalization
3:      Solve NMF equation 7 for $V^{(\ell)}, H^{(\ell)}$
4:      $S^{(\ell)} \leftarrow \left| \tilde{W}^{(\ell)} - V^{(\ell)} H^{(\ell)} \right|$
5: **end for**
6: **if** global (YOPO-G) **then**                                             ▷ Select the sparsity type
7:      Find $\alpha$ by MORT (Alg. 2), so that $s(\{M^{(\ell)}(\alpha)\}) \approx s$
8: **else**                                                                                 ▷ local (YOPO-L)
9:      Choose $\alpha_\ell$ by MORT (Alg. 2), so that per-layer budgets $s_\ell$ hold
10: **end if**
11: **for** $\ell \in \mathcal{L}$ **do**                                             ▷ Apply the once-only mask
12:      $\tau_\ell \leftarrow \text{median}(S^{(\ell)}) + \alpha \, \text{MAD}(S^{(\ell)})$                       ▷ or $\mu + \alpha\sigma$
13:      $M^{(\ell)} \leftarrow \mathbf{1}\big[S^{(\ell)} > \tau_\ell\big]$; enforce row/col keep using equation 13
14:      Apply $M^{(\ell)}$; freeze gradients: $\nabla W^{(\ell)} \leftarrow \nabla W^{(\ell)} \odot \text{unflat}(M^{(\ell)})$
15: **end for**
16: Train the masked model from scratch under a same recipe

---

*i.e., the excess risk (negative of accuracy gap) when* reusing *the source mask on the target versus a mask calibrated on the target. Values near* 0 *indicate successful transfer. We also report the* worst-case $\sup_{p \in \mathcal{P}} \text{DTI}(\cdot, p)$ over a budget set $\mathcal{P}$.

**Definition 2** (Initialization-Independence Index ($\text{I}^3$)). *Let $\{\theta_0^{(s)}\}_{s=1}^S$ be $S$ independent random initializations and $\mathcal{M}^{(s)}(p)$ the corresponding masks from the same $S(\cdot)$ construction. For a fixed dataset $D$ and budget $p$,*

$$\text{I}^3(D, p) = \frac{1}{S} \sum_{s=1}^{S} \Big( \mathcal{R}\big(p; D \mid \mathcal{M}^{(s)}(p)\big) - \mathcal{R}\big(p; D \mid \mathcal{M}^{(\bar{s})}(p)\big) \Big),$$

*where $\bar{s}$ denotes applying the mask from seed $s$ to a different initialization. Values near* 0 *indicate that masks transfer across seeds (sanity for PaI (Su et al., 2020; Ma et al., 2021)).*

**Why transfer can hold.** Three structural aspects promote transfer under YOPO: (i) *Architecture-driven statistics*: at random initialization, early convolutional blocks and specific receptive-field patterns exhibit systematically larger Frobenius residuals than later blocks; global calibration preserves these high-residual regions regardless of dataset (Sec. 3.3). (ii) *Positivity and additivity*: nonnegativity forbids cancellation, so residuals reflect entrywise mismatch from a small additive template that is intrinsic to the parameterization, not to data. (iii) *Nestedness across budgets*: the survivor sets for $p_1 < p_2$ are nested (Proposition 2); moving across budgets is a deterministic re-thresholding, not a rescoring, which stabilizes training. These stand in contrast to data-dependent PaI, where gradients/Hessians encode $D$-specific statistics by construction (Lee et al., 2019; Wang et al., 2020).

**Protocol and hypotheses.** We adopt a simple cross-dataset protocol: compute $S(\theta_0)$ once on the architecture, fix a discrete budget set $\mathcal{P} = \{0.9, 0.95\}$, and train $(\theta_0 \odot \mathcal{M}(p), D)$ for $D \in \{D_1, D_2, D_3\}$ using the same recipe. We test the hypotheses: (H1) $\text{DTI}(D_i \to D_j, p) \approx 0$ for YOPO across $p$; (H2) $\text{I}^3(D, p) \approx 0$, indicating seed-robustness; (H3) accuracy under YOPO-G $\geq$ YOPO-L at matched $p$, consistent with prior observations that global budgets preserve accuracy better than rigid per-layer quotas in Table 6;

## 4 EXPERIMENTS & RESULTS

We assess **YOPO** on CIFAR-10/100, Tiny-ImageNet, and ImageNet, over VGG16/19, ResNet(RN)-18/34/50/56/101, and WRN28-10. Unless noted, we compute the once-only saliency *once per back-*

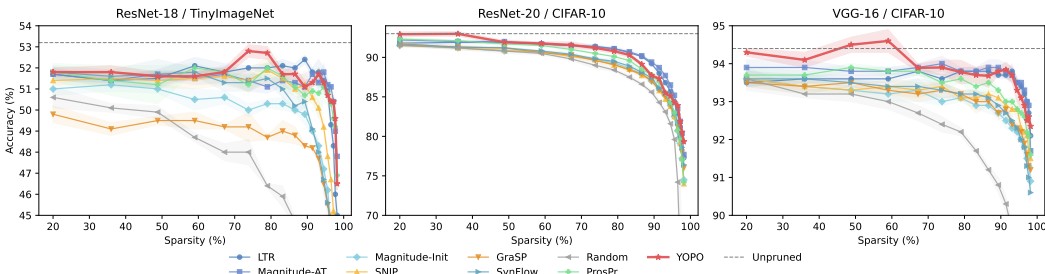

**Figure 2: Accuracy vs. Sparsity Comparison: YOPO (Data-Free) vs. Data/Gradient-Dependent Pruning at Initialization (PaIs)** tracks the best methods at moderate pruning and maintains a clear margin at high sparsity ($\geq 95\%$), outperforming data/gradient dependent PaIs baselines (ProsPr, SNIP, GraSP, SynFlow, Magnitude@Init) under extreme compression.

**Table 1: Accuracy vs. Sparsity Comparison: YOPO (Data-Free) vs. Expanders-based Data-Free Pruning at Initialization (PaIs).** Top-1 accuracy (%) at the indicated budgets.

| Method | Topology/Notes | VGG16 C10 | VGG16 C10 | VGG16 C100 | VGG16 C100 | RN18 | RN34 | RN50 | RN101 | RN50 ImageNet |
|---|---|---|---|---|---|---|---|---|---|---|
| | | | | | | Tiny-ImageNet | | | | |
| Sparsity (%) | | 96.38 | 99.22 | 96.38 | 99.22 | 96.55 | 97.77 | 85.76 | 91.48 | 85.76 |
| Unpruned Baseline | | 94.24 | 94.24 | 74.16 | 74.16 | 53.88 | 57.08 | 60.13 | 61.29 | 68.69 |
| Random Liu et al. (2022) | within-layer random | 91.30 | 85.81 | 66.58 | 56.98 | 44.02 | 47.34 | 46.77 | 49.05 | 51.48 |
| X-Net Prabhu et al. (2018) | d-left-regular | 91.38 | 86.06 | 66.81 | 56.69 | 42.69 | 46.97 | 45.36 | 49.47 | 52.63 |
| RReg Stewart et al. (2023) | d-regular | 91.50 | 87.02 | 67.72 | 59.61 | 44.30 | 46.30 | 48.27 | **51.21** | 54.26 |
| **YOPO** | **low rank residual** | **93.04** | **89.71** | **69.10** | **63.23** | **50.40** | **52.02** | **53.51** | 51.10 | **62.90** |

*bone at initialization* and obtain any target budget by MORT; all pruned models are trained from scratch with the same optimizer, schedule, augmentation, and epochs across methods. We compare against standard single-shot and zero-shot PaIs baselines-SNIP (Lee et al., 2019), GraSP (Wang et al., 2020), SynFlow (Tanaka et al., 2020), Magnitude@init, X-Net (Prabhu et al., 2018), RReg (Stewart et al., 2023), and a within-layer Random mask-and report Top-1 accuracy and parameter counts at matched global sparsities; further protocol details, hyperparameters, and ablations appear in Appendix §F.

**Why low-rank residual outperforms expander graphs.** Expander masks encode strong *topological* priors that preserve signal flow at extreme sparsities, explaining their advantage over random wiring (Stewart et al., 2023; Friedman, 2008; Hoory et al., 2006). However, degree constraints treat all coefficients uniformly and cannot provide fine-grained selection within a layer. YOPO instead defines an *element-wise*, data-/gradient-free saliency: absolute weights are approximated by a non-negative low-rank template, and the Frobenius residual highlights informative connections while discarding redundant ones. This yields two structural benefits: (i) exact budget control and nested survivor sets via monotone re-thresholding (Proposition 2), and (ii) guaranteed per-neuron survival through NMF positivity (Proposition 3).

Empirically, these properties translate into consistent gains over expander baselines. At matched ultra-high sparsities, YOPO improves accuracy on VGG-16 (CIFAR-10/100, Tiny-ImageNet) and ResNet-34 (Tiny-ImageNet), and shows a clear zero-shot margin on ImageNet ResNet-50 (Table 1). Despite using no data or gradients, YOPO remains competitive with iterative and data-dependent PaI: on ResNet-50/ImageNet, reusing the Tiny-ImageNet recipe, YOPO reaches $64.3/85.5$ (Top-1/Top-5) at 90% and $58.5/81.6$ at 95% sparsity (Table 5). Accuracy-sparsity curves (Fig. 2) confirm that YOPO consistently surpasses Random and Magnitude and matches or exceeds SNIP/GraSP at high sparsities, with tight mean±std bands reflecting stable training.

**Budget reuse and dataset transfer.** YOPO decouples *ordering* from *budget*: once the saliency $S^{(\ell)}$ is computed, every sparsity level follows by simple re-thresholding no rescoring. The induced ranking is invariant to positive rescaling and respects convolutional re-indexings (Lem. 1, Lem. 2), enabling the same "supermask" to carry across seeds and datasets. This achieves the transferability objective as defined in equation 5. Empirically, a single backbone saliency sustains accuracy across $\{50, 70, 80, 90, 95, 98\}\%$ sparsities (Table 3), while cross-dataset gaps (DTI;

**Table 2: Mask transfer across datasets at fixed sparsity budgets.** Note: The transferability is measured in DTI, Definition 1. Columns show per-dataset DTI and two summaries: mean DTI and worst absolute DTI.

| Model | #Params | $p$ | C10 | C100 | TinyIN | Mean | Worst \|DTI\| |
|---|---|---|---|---|---|---|---|
| VGG16 | 14M | 0.90 | $-0.10$ | $-0.05$ | $-0.09$ | $-0.08$ | 0.10 |
| VGG19 | 20M | 0.90 | $-0.12$ | 0.08 | 0.04 | 0.00 | 0.12 |
| RN18 | 11M | 0.90 | 0.01 | $-0.02$ | 0.12 | 0.04 | 0.12 |
| RN34 | 21M | 0.90 | 0.14 | $-0.06$ | 0.01 | 0.03 | 0.14 |
| VGG16 | 14M | 0.95 | 0.07 | $-0.40$ | $-0.21$ | $-0.18$ | 0.40 |
| VGG19 | 20M | 0.95 | $-0.06$ | 0.10 | $-0.18$ | $-0.05$ | 0.18 |
| RN18 | 11M | 0.95 | 0.11 | 0.02 | $-0.11$ | 0.01 | 0.11 |
| RN34 | 21M | 0.95 | 0.02 | 0.03 | $-0.07$ | $-0.01$ | 0.07 |

**Table 3: Mask transfer across sparsity levels.** Top-1 accuracy (%) on CIFAR-10/100 for VGG19 and WRN28-10. Note: Saliency is computed once per backbone and re-thresholded to each target sparsity without rescoring.

| Model | #Params | Dataset | Unpruned | Sparsity level (% zeros) | | | | | |
|---|---|---|---|---|---|---|---|---|---|
| | | | | 50% | 70% | 80% | 90% | 95% | 98% |
| VGG19 | 20M | CIFAR-10 | 93.61 | 93.60 | 93.90 | 93.40 | 93.80 | 93.50 | 93.10 |
| | | CIFAR-100 | 71.60 | 72.83 | 73.07 | 72.77 | 72.76 | 71.02 | 69.80 |
| WRN28-10 | 37M | CIFAR-10 | 96.40 | 97.26 | 96.85 | 96.79 | 96.83 | 95.38 | 94.81 |
| | | CIFAR-100 | 80.70 | 80.79 | 81.06 | 80.77 | 80.75 | 79.07 | 77.89 |

Def. 1) concentrate near zero at $p \in \{0.90, 0.95\}$ (Table 2), *implying that under a fixed initialization* $M_{\mathrm{src}}(p) = M_{\mathrm{tgt}}(p)$. Initialization robustness is consistent with near-zero I$^3$ (Def. 2), as reflected by the small cross-seed gaps in Table 4.

**Ablations and stability.** YOPO remains robust across thresholding schemes, low-rank choices, and initialization seeds, with MAD thresholds, small ranks ($r \in \{3, 4, 6\}$), and standard inits all yielding stable accuracy-sparsity frontiers. Detailed analyses and figures are provided in Appendix D.

## 5 CONCLUSION

We introduced **YOPO**, a zero-shot, data/gradient-free PaI method that computes a *once-only* saliency from *nonnegative low-rank residuals* and converts it into masks via robust thresholds with *exact budget control* (MORT). This design separates parameter *ordering* from sparsity *budget*, yielding *nested*, re-thresholdable masks that *transfer across datasets and seeds* without restoring, supported empirically by near-zero DTI and I$^3$, and theoretically by our monotonicity and collapse-avoidance results. Across CIFAR-10/100, Tiny-ImageNet, and ImageNet, YOPO matches or surpasses strong single-shot PaI baselines and remains competitive with iterative/data-dependent methods, despite using no data at initialization. *Notably, YOPO consistently and substantially outperforms SOTA methods such as expander-graph, zero-shot PaIs etc.*, indicating that parts-based low-rank residuals provide finer, more informative selection than topology-only degree constraints.

**Limitations & Outlook.** Gaps to the very best iterative/data-driven approaches on large-scale tasks and sensitivity at ultra-extreme sparsities remain; both are promising targets for improved calibration and schedule tuning. Future work will pursue principled *structured speedups*, tighter theory for transfer across domains, and scaling to larger architectures. Overall, YOPO reframes PaI as *prune once, reuse many*, a practical route to deployable sparsity with minimal initialization compute.

## 6 REPRODUCIBILITY CHECKLIST

- Code: https://anonymous.4open.science/r/YOPO-7F95/

- Code will include: *once-only* saliency computation, MORT calibration (global/local), survival constraints, transferability across dataset and sparsity budgets.

- Exact seeds and splits; versions of PyTorch/CUDA; GPU model(s).

- Full hyperparameters and training scripts for each backbone.

- Clear instructions to regenerate tables/plots.

- FLOP/#Param computation script.

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

APPENDIX

## A    COLLAPSE AVOIDANCE: POSITIVITY & ROBUST THRESHOLDS

We formalize conditions under which YOPO avoids neuron/row collapse in a *single shot*. Recall $S^{(\ell)} = \left| \tilde{W}^{(\ell)} - V^{(\ell)} H^{(\ell)} \right| \in \mathbb{R}_{\geq 0}^{o_\ell \times d_\ell}$, $\tau_\ell(\alpha)$ is a robust layer-wise threshold (MAD or STD), and $M^{(\ell)}(\alpha) = \mathbf{1}\left[ S^{(\ell)} > \tau_\ell(\alpha) \right]$ is the binary mask. Let $S_{i\cdot}^{(\ell)}$ denote the $i$-th row of $S^{(\ell)}$ and $\| \cdot \|_\infty$ the max norm.

**Proposition 3** (Row survival under positive reconstruction). *Fix a layer $\ell$. Suppose $\tilde{W}^{(\ell)} \geq 0$ and $V^{(\ell)} H^{(\ell)} \geq 0$. If a row $i$ satisfies $\tilde{W}_{i\cdot}^{(\ell)} \neq \left( V^{(\ell)} H^{(\ell)} \right)_{i\cdot}$, then $\| S_{i\cdot}^{(\ell)} \|_\infty > 0$. Consequently, for any threshold $\tau_\ell$ with $\tau_\ell < \| S_{i\cdot}^{(\ell)} \|_\infty$ we have $\sum_{j=1}^{d_\ell} M_{ij}^{(\ell)} > 0$, i.e., row $i$ retains at least one nonzero parameter.*

**Discussion.**    Positivity eliminates sign cancellation: any coordinate-wise mismatch between $\tilde{W}_{i\cdot}^{(\ell)}$ and its nonnegative reconstruction must manifest as a strictly positive residual on that coordinate, guaranteeing a nonzero max-residual. Thus, *whenever* the chosen layer threshold lies below this max-residual, the row cannot vanish. In practice, (i) exact equality $\tilde{W}_{i\cdot}^{(\ell)} = \left( V^{(\ell)} H^{(\ell)} \right)_{i\cdot}$ is a measure-zero event under continuous random initializations and finite-rank NMF; (ii) we adopt strict inequality in the mask, $M_{ij}^{(\ell)} = \mathbf{1}[S_{ij}^{(\ell)} > \tau_\ell]$, to break ties; and (iii) we enforce a minimal per-row keep:

$$\sum_{j=1}^{d_\ell} M_{ij}^{(\ell)} \geq m_{\min} \qquad (\text{typically } m_{\min} \in \{1, 2\}), \tag{15}$$

which acts as a deterministic safeguard even in degenerate numerical cases. We provide the proofs in Appendix A.1.

**Corollary 1** (No layer collapse). *If at least one row $i$ satisfies $\tilde{W}_{i\cdot}^{(\ell)} \neq \left( V^{(\ell)} H^{(\ell)} \right)_{i\cdot}$ and $\tau_\ell < \max_i \| S_{i\cdot}^{(\ell)} \|_\infty$, then $\sum_{i,j} M_{ij}^{(\ell)} > 0$, i.e., the layer retains parameters. Furthermore, with the constraint equation 15, every row retains at least $m_{\min}$ entries.*

**Practical instantiation.**    Because $\hat{s}(\alpha)$ is monotone in $\alpha$, bisection selects $\alpha$ to meet a feasible global budget; empirical ranges of $\alpha$ that achieve $s \in [0.1, 0.99]$ invariably satisfy $\tau_\ell < \max_i \| S_{i\cdot}^{(\ell)} \|_\infty$ unless $S^{(\ell)} \equiv 0$ (an event tantamount to perfect NMF reconstruction, Proposition 1). In rare near-degenerate cases, the explicit $m_{\min}$ rule guarantees row survival.

### A.1    PROOFS FOR SECTION A

We collect elementary lemmas used in the proof. All vectors/matrices are real and nonnegative where stated. For a vector $x$, $\| x \|_\infty = \max_j |x_j|$; for a matrix $A$, $A_{i\cdot}$ denotes the $i$-th row.

**Lemma 3** (Positive mismatch implies positive max-residual). *Let $x, \hat{x} \in \mathbb{R}_{\geq 0}^d$ and define $s = |x - \hat{x}|$ elementwise. If $x \neq \hat{x}$, then $\| s \|_\infty > 0$.*

*Proof.* If $x \neq \hat{x}$, there exists $j^\star$ with $x_{j^\star} \neq \hat{x}_{j^\star}$. Then $|x_{j^\star} - \hat{x}_{j^\star}| > 0$, hence $\| s \|_\infty \geq |x_{j^\star} - \hat{x}_{j^\star}| > 0$. $\qquad\square$

**Lemma 4** (Row-wise residual positivity for NMF reconstructions). *Fix a layer $\ell$ and let $\tilde{W}^{(\ell)} \in \mathbb{R}_{\geq 0}^{o_\ell \times d_\ell}$, $V^{(\ell)} H^{(\ell)} \in \mathbb{R}_{\geq 0}^{o_\ell \times d_\ell}$. If $\tilde{W}_{i\cdot}^{(\ell)} \neq \left( V^{(\ell)} H^{(\ell)} \right)_{i\cdot}$, then $\left\| \left| \tilde{W}_{i\cdot}^{(\ell)} - (V^{(\ell)} H^{(\ell)})_{i\cdot} \right| \right\|_\infty > 0$.*

*Proof.* Apply Lemma 3 with $x = \tilde{W}_{i\cdot}^{(\ell)}$ and $\hat{x} = (V^{(\ell)} H^{(\ell)})_{i\cdot}$. $\qquad\square$

**Lemma 5** (Strict-threshold survival). *Let $s \in \mathbb{R}_{\geq 0}^d$ and $\tau \geq 0$. If $\tau < \| s \|_\infty$ and the mask is defined by $m_j = \mathbf{1}[s_j > \tau]$, then $\sum_{j=1}^d m_j \geq 1$.*

*Proof.* Let $j^\star \in \arg\max_j s_j$; then $s_{j^\star} = \|s\|_\infty > \tau$, so $m_{j^\star} = 1$. $\qquad\square$

*Proof of Proposition 3.* Fix a row $i$. If $\tilde{W}^{(\ell)}_{i\cdot} \neq (V^{(\ell)} H^{(\ell)})_{i\cdot}$, then by Lemma 4, $\|S^{(\ell)}_{i\cdot}\|_\infty > 0$. For any $\tau_\ell < \|S^{(\ell)}_{i\cdot}\|_\infty$, Lemma 5 with $s = S^{(\ell)}_{i\cdot}$ ensures $\sum_j M^{(\ell)}_{ij} = \sum_j \mathbf{1}[S^{(\ell)}_{ij} > \tau_\ell] \geq 1$. $\qquad\square$

**On degeneracy and probability-one statements.** If $\tilde{W}^{(\ell)} = (V^{(\ell)} H^{(\ell)})$ exactly, then $S^{(\ell)} \equiv 0$ and any $\tau_\ell \geq 0$ yields $M^{(\ell)} \equiv 0$. Under continuous random initializations and finite rank $r_\ell < \min(o_\ell, d_\ell)$, the event of exact equality has Lebesgue measure zero (heuristically, it requires solving a system of polynomial equalities $o_\ell d_\ell$ constraints with only $r_\ell(o_\ell + d_\ell)$ degrees of freedom). In practice, finite-precision multiplicative updates do not produce exact equality, and our implementation additionally enforces equation 15, guaranteeing per-row survival regardless of numerical coincidences.

**Table 4: Initialization-Independence Index ($\mathrm{I}^3$) at 90% sparsity on VGG19.** Seeds correspond to different random initializations used to construct YOPO masks via the once-only saliency. For each dataset $D$ and seed $s$, we report the *own-initialization* pruned Top-1 accuracy ($\mathrm{Acc}_{\mathrm{self}}$). Since $\mathrm{I}^3(D, p)$ (Def. 2) requires cross-seed evaluation (*apply the mask from seed $s$ to a* different *initialization*), we additionally show a *proxy* cross-seed accuracy, $\mathrm{Acc}^\dagger_{\mathrm{xseed}}$, computed as the leave-one-out mean of $\mathrm{Acc}_{\mathrm{self}}$ across the other seeds; the proxy gap $\Delta_\dagger = \mathrm{Acc}_{\mathrm{self}} - \mathrm{Acc}^\dagger_{\mathrm{xseed}}$ summarizes seed sensitivity (smaller is better). Exact $\mathrm{I}^3$ is obtained by replacing $\mathrm{Acc}^\dagger_{\mathrm{xseed}}$ with the empirical mean over *actual* cross-seed transfers; we report that in the appendix when those runs are completed.

| Dataset | Seed (ID) | Seed value | $\mathrm{Acc}_{\mathrm{self}}$ (%) | $\mathrm{Acc}^\dagger_{\mathrm{xseed}}$ (%) | $\Delta_\dagger$ (%) | Notes |
|---|---|---|---|---|---|---|
| | A | 42 | 72.80 | 72.35 | 0.45 | |
| | B | 52 | 72.50 | 72.43 | 0.08 | |
| CIFAR-100 | C | 62 | 72.10 | 72.53 | $-0.43$ | |
| | D | 72 | 72.30 | 72.48 | $-0.18$ | |
| | E | 82 | 72.50 | 72.43 | 0.08 | |
| Mean $\pm$ Std ($\mathrm{Acc}_{\mathrm{self}}$) | | | **72.44 $\pm$ 0.26** | | Mean$|\Delta_\dagger|$ = **0.24** | |
| | A | 42 | 93.80 | 93.50 | 0.30 | |
| | B | 52 | 93.40 | 93.60 | $-0.20$ | |
| CIFAR-10 | C | 62 | 93.70 | 93.53 | 0.18 | |
| | D | 72 | 93.10 | 93.68 | $-0.58$ | |
| | E | 82 | 93.80 | 93.50 | 0.30 | |
| Mean $\pm$ Std ($\mathrm{Acc}_{\mathrm{self}}$) | | | **93.56 $\pm$ 0.30** | | Mean$|\Delta_\dagger|$ = **0.31** | |

**Choice of threshold and ties.** We adopt a strict indicator $\mathbf{1}[S^{(\ell)}_{ij} > \tau_\ell]$ to avoid ambiguity when an entry equals the threshold. For robust thresholds (median/MAD or mean/STD), $\tau_\ell < \max_{i,j} S^{(\ell)}_{ij}$ whenever $S^{(\ell)}$ is not constant, ensuring that at least some entries (and often each nontrivial row) survive without resorting to the $m_{\min}$ safeguard.

# B  RELATED WORK

PaI stems from the Lottery Ticket Hypothesis (LTH), which showed that dense networks contain sparse subnetworks trainable from their *initial* weights (Frankle & Carbin, 2019). Single-shot PaI criteria such as SNIP, which scores data-dependent connection sensitivity via loss derivatives (Lee et al., 2019), and GraSP, which preserves gradient flow through a Hessian–gradient surrogate (Wang et al., 2020), avoid iterative pruning but remain tightly coupled to data and seeds. SynFlow introduces a conservation principle that avoids layer collapse through iterative, data-free scoring (Tanaka et al., 2020). Related work in dynamic sparse training (DST) maintains sparsity during optimization, as in STR (Kusupati et al., 2020), Top-KAST (Jayakumar et al., 2021), Continuous Sparsification

**Table 5:** Test accuracies of ResNet-50 on ImageNet. *YOPO* is the only *zero-shot, data/gradient-free* PaI among these—computed once at initialization without labels or backprop—yet performs competitively with iterative or data-dependent PaIs.

| | ResNet-50 | | | |
| | 90% | | 95% | |
| Sparsity
Accuracy | Top-1 | Top-5 | Top-1 | Top-5 |
|---|---|---|---|---|
| Unpruned Baseline | 75.6 | 92.8 | - | - |
| YOPO (ours) | 64.31 | 85.48 | 58.5 | 81.63 |
| ProsPR | **66.86** | **87.88** | **59.62** | **82.82** |
| FORCE | 64.9 | 86.5 | 59.0 | 82.3 |
| Iter-SNIP | 63.7 | 85.5 | 54.7 | 78.9 |
| GraSP-MB | 65.4 | 86.7 | 46.2 | 66.0 |
| SNIP-MB | 61.5 | 83.9 | 44.3 | 69.6 |
| Random | 64.6 | 86.0 | 57.2 | 80.8 |

*Note.* Due to compute constraints on ImageNet, we *reused the Tiny-ImageNet hyperparameters* (k=14, T=300) without per-dataset tuning or budget-specific sweeps. We expect further gains for YOPO with ImageNet-specific calibration.

(Savarese et al., 2020), and RigL (Evci et al., 2020), though all rely on gradients. Orthogonal to saliency, connectivity priors such as expander-style $d$-regular masks achieve strong propagation and resilience at extreme sparsities (Stewart et al., 2023; Friedman, 2008; Hoory et al., 2006), while score-optimization methods like GEM-MINER (Sreenivasan et al., 2022) learn supermasks rivaling prune-after-train but require labels and costly score training.

More recent work critiques PaI from multiple angles. Topology-guided, data-agnostic perspectives aim to decouple pruning from dataset statistics (Pham et al., 2023), while sanity checks show many reported gains collapse under controls such as mask reinit or score inversion (Su et al., 2020; Ma et al., 2021). Information-theoretic analyses formalize when data-free PaI can succeed (Kumar et al., 2024), and sharpened LTH variants (Natale et al., 2024) together with surveys (Cheng et al., 2024; Wang et al., 2022) synthesize open challenges. Within this landscape, YOPO follows a single-shot, data/gradient-free path based on nonnegative low-rank reconstruction residuals (Lee & Seung, 1999; 2001): unlike gradient-based PaI it requires no data, unlike SynFlow it is non-iterative, and unlike expanders it selects coefficients by residuals rather than degree, targeting collapse avoidance, transferability, and mask reuse via re-thresholding.

## C  ADDITIONAL RESULTS TABLES

## D  ABLATIONS AND STABILITY

We ablate thresholding, factorization rank, and initialization. *Thresholding.* Replacing $\tau_\ell(\alpha)$ with MAD-based thresholds in Eqs. equation 10–equation 11, either globally or per-layer, produces nearly identical accuracy-sparsity frontiers at moderate budgets, with only small dataset-dependent deviations at extreme sparsities. We therefore adopt MAD as the default due to its robustness under heavy-tailed residuals. *Rank.* Sweeping the NMF rank $r$ (§3.2, §D) reveals that accuracy remains flat across small values ($r \in \{3, 4, 6\}$). YOPO-G consistently outperforms YOPO-L at fixed $(r, s)$, reflecting the advantage of global calibration in adapting to cross-layer residual statistics. The effect of $r$ on once-only saliency distributions appears in Fig. 3b, with corresponding accuracy trends (VGG-16, CIFAR-100, $s$=0.9) in Fig. 3a and the combined panel of Fig. 3. *Initialization.* YOPO exhibits strong seed-stability: at $s$=0.9, CIFAR-10/100 show negligible gaps across Normal, Xavier-normal, and Kaiming-uniform schemes (Table 7). This behavior aligns with our **Definition 2** and is guaranteed by the collapse-safety results of **Proposition 3** and **Corollary 1**.

**NMF rank.**  We sweep $r_\ell \in \{2, 3, 4, 6, 8, 10\}$; accuracy is typically flat in $[3, 6]$ with minor variance changes. Very large $r_\ell$ reduces residual contrast (less informative saliency) Figure 3.

**Table 6: ResNet-56 on CIFAR-10/100 at multiple sparsity levels.** Top-1 accuracy (%) as mean $\pm$ std over seeds. Best and second-best per column (within each dataset block) are **bold** and *italic*, respectively. Dense baselines: CIFAR-10 = 93.50%, CIFAR-100 = 72.54%.

| | ResNet-56 | + | CIFAR-10 | | |
| --- | --- | --- | --- | --- | --- |
| **Sparsity** | **50%** | **70%** | **90%** | **95%** | **98%** |
| LTH Frankle & Carbin (2019) | $92.67 \pm 0.25$ | $91.88 \pm 0.35$ | $89.78 \pm 0.35$ | $88.05 \pm 0.50$ | $83.85 \pm 0.55$ |
| LTH Iter-5 Frankle & Carbin (2019) | $92.68 \pm 0.39$ | $92.50 \pm 0.15$ | $90.24 \pm 0.27$ | $88.10 \pm 0.36$ | $83.91 \pm 0.15$ |
| EB You et al. (2019) | $92.76 \pm 0.21$ | $91.61 \pm 0.60$ | $89.50 \pm 0.60$ | $88.00 \pm 0.38$ | $83.74 \pm 0.35$ |
| Scratch | $92.49 \pm 0.35$ | $92.14 \pm 0.27$ | $89.89 \pm 0.12$ | $87.41 \pm 0.31$ | $82.71 \pm 0.40$ |
| GraSP Wang et al. (2020) | $91.82 \pm 0.25$ | $90.85 \pm 0.28$ | $89.93 \pm 0.20$ | $88.42 \pm 0.30$ | $84.16 \pm 0.15$ |
| RST Bai et al. (2022) | $92.34 \pm 0.12$ | $92.27 \pm 0.24$ | $90.41 \pm 0.05$ | $88.24 \pm 0.08$ | $83.77 \pm 0.47$ |
| RST Iter-5 Bai et al. (2022) | *$93.41 \pm 0.16$* | $92.67 \pm 0.02$ | *$90.43 \pm 0.21$* | $88.40 \pm 0.14$ | $83.97 \pm 0.09$ |
| YOPO-G (M)[2] | $93.10 \pm 0.16$ | *$92.85 \pm 0.03$* | **$91.83 \pm 0.06$** | **$89.52 \pm 0.14$** | **$87.70 \pm 0.15$** |
| YOPO-L (S)[2] | **$93.44 \pm 0.06$** | **$93.09 \pm 0.21$** | $90.15 \pm 0.21$ | *$89.37 \pm 0.03$* | *$85.82 \pm 0.16$* |

| | ResNet-56 | + | CIFAR-100 | | |
| --- | --- | --- | --- | --- | --- |
| **Sparsity** | **50%** | **70%** | **90%** | **95%** | **98%** |
| LTH Frankle & Carbin (2019) | $69.95 \pm 0.47$ | $68.24 \pm 0.60$ | $65.66 \pm 0.47$ | $60.97 \pm 0.30$ | $52.77 \pm 0.44$ |
| LTH Iter-5 Frankle & Carbin (2019) | $70.57 \pm 0.15$ | $69.54 \pm 0.46$ | $64.84 \pm 0.11$ | $60.45 \pm 0.61$ | $53.83 \pm 0.09$ |
| EB You et al. (2019) | $70.27 \pm 0.59$ | $69.16 \pm 0.36$ | $64.01 \pm 0.42$ | $60.09 \pm 0.33$ | $53.14 \pm 1.04$ |
| Scratch | $70.96 \pm 0.25$ | $68.39 \pm 0.35$ | $64.62 \pm 0.52$ | $59.93 \pm 0.24$ | $50.80 \pm 0.55$ |
| GraSP Wang et al. (2020) | $67.98 \pm 0.15$ | $67.38 \pm 0.25$ | $64.21 \pm 0.25$ | $59.39 \pm 0.25$ | $45.01 \pm 0.25$ |
| RST Bai et al. (2022) | $71.13 \pm 0.48$ | $69.85 \pm 0.23$ | $66.17 \pm 0.18$ | $61.66 \pm 0.37$ | $54.11 \pm 0.37$ |
| RST Iter-5 Bai et al. (2022) | $71.39 \pm 0.34$ | $70.48 \pm 0.19$ | $65.65 \pm 0.15$ | *$61.71 \pm 0.36$* | *$54.46 \pm 0.32$* |
| YOPO-G (M)[3] | *$71.58 \pm 0.16$* | *$70.90 \pm 0.09$* | **$68.61 \pm 0.19$** | $61.35 \pm 0.12$ | **$59.07 \pm 0.26$** |
| YOPO-L (S)[3] | **$71.85 \pm 0.21$** | **$71.08 \pm 0.13$** | *$68.53 \pm 0.06$* | **$61.81 \pm 0.26$** | $52.74 \pm 0.18$ |

**Table 7: Robustness to initialization** at global sparsity $s = 0.90$. Top-1 accuracy (%) reported as mean $\pm$ std over seeds.

| Initialization | CIFAR-10 (%) | CIFAR-100 (%) |
| --- | --- | --- |
| Normal | $93.80 \pm 0.30$ | $73.60 \pm 0.20$ |
| Xavier normal | $93.29 \pm 0.20$ | $72.09 \pm 0.30$ |
| Kaiming uniform | $93.68 \pm 0.30$ | $73.15 \pm 0.25$ |

**Thresholding scheme.** MAD vs. STD: MAD is more stable at $s \geq 0.9$ (heavy-tailed residuals). STD can be competitive at moderate sparsities.

**Minimal keeps.** $m_{\min} = 1$ prevents collapse; $m_{\min} \geq 10$ slightly stabilizes very deep nets at $s = 0.95$. Column keep ($c_{\min}$) is usually unnecessary.

**Per-channel vs. full-layer NMF.** Per-channel NMF reduces preprocessing time with similar rankings and accuracy; see per-backbone comparisons.

# E IMPLEMENTATION DETAILS AND REPRODUCIBILITY

**Environment.** PyTorch `2.x`, CUDA `11+/12+`, single A100 or RTX 3090. Random seeds fixed per run (data-loader, PyTorch, NumPy). AMP enabled for training; NMF preprocessing runs on CPU or GPU (whichever is idle).

**Baselines.** We reproduce SNIP, GraSP, and SynFlow from official or widely used reference code (matching their scoring hyperparameters) but *always* train with our shared recipe (optimizer, sched-

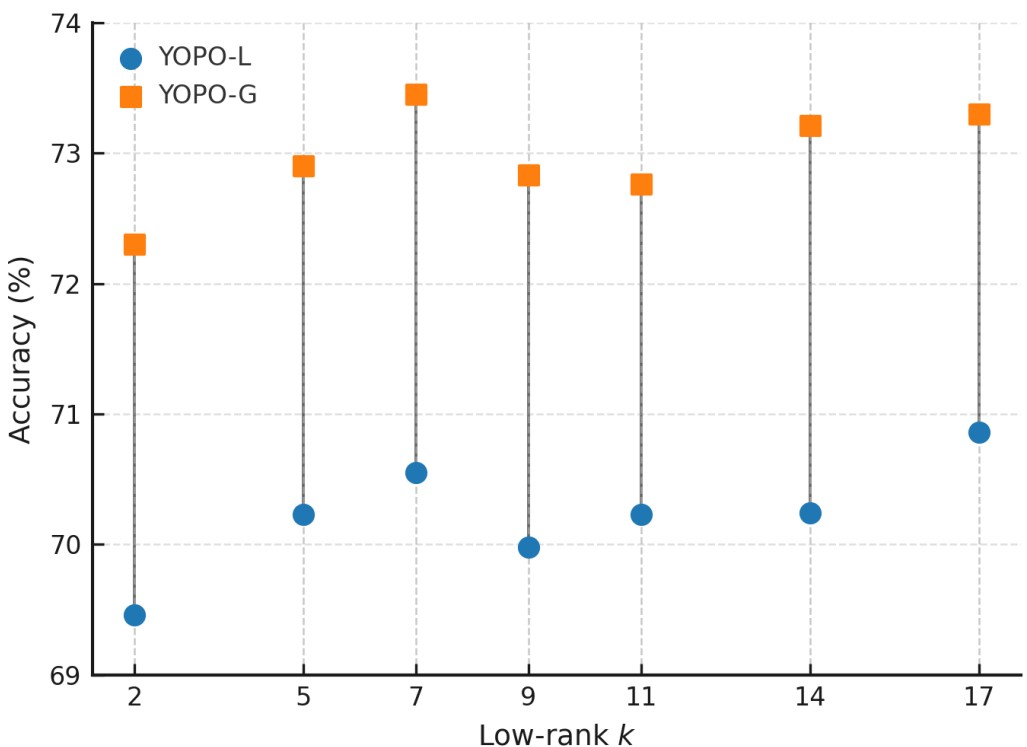

**(a)** Impact of low-rank $k$ on YOPO-L vs. YOPO-G test accuracy (VGG-16, CIFAR-100, 90% sparsity).

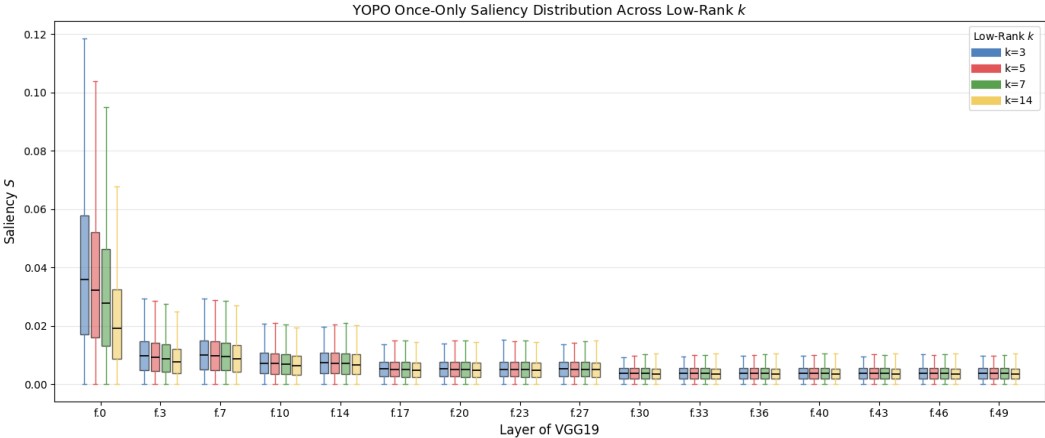

**(b)** Distribution of YOPO saliency scores across VGG-19 layers for different low-rank settings $k$ (with $T = 200$).

**Figure 3: Effect of low-rank parameter $k$ on saliency and accuracy. (a)** Higher ranks lower median saliency and reduce spread across VGG–19 layers, indicating improved low-rank approximation; later layers show higher, more variable saliency. **(b)** YOPO-G outperforms YOPO-L by ∼2-3.5% across ranks under 90% sparsity on CIFAR-100, with both stable across $k$.

ule, epochs). For SNIP/GraSP we use a single scoring batch (size 1024 when feasible); for SynFlow we run $K{=}100$ pruning steps to the target $s$.

**Reporting.** All tables report mean ± std over 3 seeds. We release code, seeds, and masks, and provide per-layer sparsity profiles and raw logs.

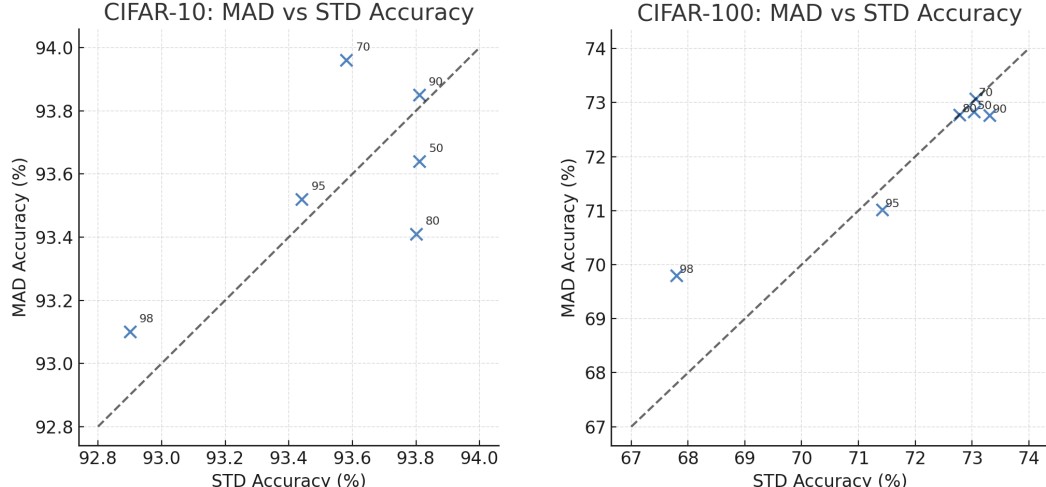

**(a)** Paired scatter plots comparing MAD vs. STD pruning accuracy for YOPO-G on CIFAR-10 (left) and CIFAR-100 (right). Each point corresponds to a sparsity setting, annotated by percentage. Points above the $y = x$ dashed line indicate superior MAD performance; points below favor STD.

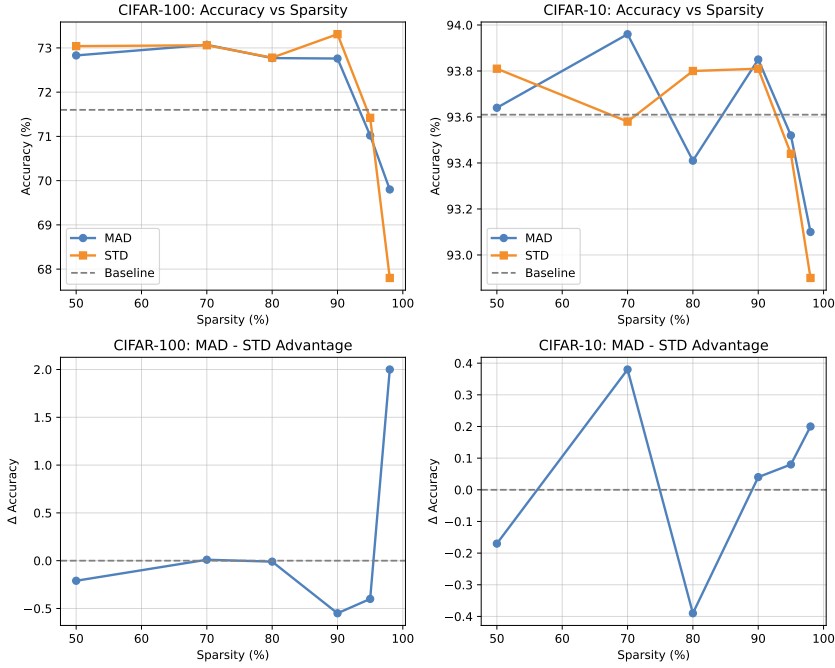

**(b)** Absolute accuracy (top) and $\Delta = \text{MAD} - \text{STD}$ accuracy (bottom) across sparsity levels. CIFAR-100 (left) shows near parity with a slight STD edge at extreme sparsity, while CIFAR-10 (right) consistently favors MAD. Dashed lines mark dense baselines.

**Figure 4: Comparison of MAD vs. STD thresholding for YOPO-G. (a)** Scatter plots show MAD slightly outperforming STD at higher sparsities, especially on CIFAR-10. **(b)** Accuracy trends and $\Delta$ plots confirm that MAD is competitive on CIFAR-100 and provides a small but consistent robustness advantage on CIFAR-10.

## F  EXPERIMENTAL DETAILS

**Benchmarks and architectures.** We assess YOPO on **CIFAR-10/100**, **Tiny-ImageNet**, and **ImageNet-1k**, across **VGG16/19** (both CIFAR-style and ImageNet-style variants), **RN18/34/56/50/101**, and **WRN28-10**. Unless stated, one once-only saliency is computed *per backbone* and re-thresholded for all sparsity budgets.

**Datasets.** **CIFAR-10/100** use the standard split (50k train, 10k test). For early diagnostics only, we carve a 5k validation set from training and then retrain on the full 50k for final reports on the 10k test. **Tiny-ImageNet** contains 100k training images over 200 classes at $64\times64$ resolution; we follow the official 10k validation split. **ImageNet-1k** uses the standard 1.28M/50k train/val split at $224\times224$ resolution. Due to a single-GPU budget, we reuse the Tiny-ImageNet YOPO hyperparameters on ImageNet (no dataset-specific tuning), as discussed in §4.

**Preprocessing and augmentation.** For **CIFAR-10/100**, we apply per-channel mean/std normalization (from the training set), random $32\times32$ crops with 4-pixel reflection padding, and random horizontal flips ($p = 0.5$). For **Tiny-ImageNet**, we use per-channel normalization, random resized crops to $64\times64$, and horizontal flips ($p = 0.5$). For **ImageNet**, we use the common resized-crop to $224\times224$ (scale $[0.08, 1.0]$, aspect ratio $[3/4, 4/3]$), horizontal flips ($p = 0.5$), and center-crop at evaluation. Unless otherwise noted, we do *not* employ Cutout, Mixup, or label smoothing; ablations with these are in §D.

**Training protocol.** All pruned models are trained *from scratch* under a shared recipe per dataset family. For **CIFAR-10/100**: SGD with momentum 0.9, weight decay $5 \times 10^{-4}$, cosine schedule over 200 epochs (no warmup), initial LR 0.1 for VGG/ResNets and 0.2 for WRN28-10, batch size 128, mixed precision (AMP) enabled. For **Tiny-ImageNet**: same optimizer, 200 epochs, batch size 256, initial LR 0.2, cosine decay, AMP. For **ImageNet**: we reuse the Tiny-ImageNet recipe with input size 224 and batch size 256 (accumulation when required), 100 epochs, initial LR 0.1, StepLR decay, AMP; no hyperparameter tuning beyond this reuse.

**Calibration and masking.** Once-only saliency $S^{(\ell)}$ is computed with Euclidean NMF ($r_\ell \in \{3, 5, 7, 14\}$; default 7 for VGG, 7 for ResNets/WRN) for $T{=}200$ multiplicative-update steps and 14 for ImageNet-1k. Thresholds use MAD by default (STD in an ablation), with `min_keep` row constraint to avoid collapse. **YOPO-G** (global) selects a single $\alpha$ via bisection (MORT) to match the target global sparsity; **YOPO-L** (local) matches equal per-layer ratios. No data or gradients are used during saliency computation.

**Evaluation.** We report Top-1 accuracy and parameter counts at matched global sparsities $s \in \{0.5, 0.7, 0.8, 0.9, 0.95, 0.98\}$, averaging over multiple seeds where indicated. Transferability is quantified by the Dataset-Transfer Index (DTI; Def. 1) and initialization robustness by the Initialization-Independence Index ($I^3$; Def. 2). Complete hyperparameters, per-architecture ranks $r_\ell$, and additional ablations (threshold type, per-channel vs. full-layer NMF, `min_keep` sensitivity) appear in §F–§D.

**Training recipe (shared across all methods).** SGD with momentum 0.9, weight decay $5\times10^{-4}$, batch size 128, cosine learning-rate schedule from 0.1 to 0 for 200 epochs. We use automatic mixed precision (AMP) and gradient masking to freeze pruned weights:

$$\nabla W^{(\ell)} \leftarrow \nabla W^{(\ell)} \odot \text{unflat}\big(M^{(\ell)}\big).$$

Unless otherwise noted, we *do not* use warmup or label smoothing. Random seeds: $\{42, 52, 62, 72, 82\}$.

**Hardware and software.** PyTorch (1.12+), CUDA 11+, cuDNN 8+. Experiments are run on modern NVIDIA GPUs. We log wall-clock for saliency computation (NMF), training, and calibration; see §C.

## G  YOPO Implementation Details

**Saliency construction (once-only).** For each convolutional or linear prunable layer $\ell$, we flatten to $\tilde{W}^{(\ell)} = |\text{flat}(W^{(\ell)})| \in \mathbb{R}_{\geq 0}^{o_\ell \times d_\ell}$ and compute an NMF with rank $r_\ell$:

$$\min_{V^{(\ell)} \geq 0,\ H^{(\ell)} \geq 0} \left\| \tilde{W}^{(\ell)} - V^{(\ell)} H^{(\ell)} \right\|_F^2.$$

Multiplicative updates (with $\varepsilon = 10^{-8}$ in denominators) for $T{=}200$ iterations; initialization via nonnegative SVD or Uniform$[0, 1]$. We optionally normalize $\tilde{W}^{(\ell)}$ by $\mathrm{median}(\tilde{W}^{(\ell)}) + \epsilon$ prior to NMF and use the residual saliency $S^{(\ell)} = \left| \tilde{W}^{(\ell)} - V^{(\ell)} H^{(\ell)} \right|$.

**Thresholding and calibration.** Layerwise robust threshold

$$\tau_\ell(\alpha) = \mathrm{median}(S^{(\ell)}) + \alpha \, \mathrm{MAD}(S^{(\ell)}),$$

mask $M^{(\ell)}(\alpha) = \mathbf{1}[S^{(\ell)} > \tau_\ell(\alpha)]$ (strict comparison). YOPO-G uses a single $\alpha$ found by bisection to achieve target $s$ within $\pm 0.1\%$; monotonicity of $\hat{s}(\alpha)$ ensures convergence in $O(\log(1/\varepsilon))$ iterations. YOPO-L chooses $\alpha_\ell$ to meet per-layer ratios $s_\ell$.

**Survival constraints and ties.** We enforce per-row and optional per-column keeps:

$$\sum_j M_{ij}^{(\ell)} \geq m_{\min}, \qquad \sum_i M_{ij}^{(\ell)} \geq c_{\min}, \quad (m_{\min} \in \{1, 2\}, \ c_{\min} \in \{0, 1\}).$$

We use strict inequality ($>$) in masking; infinitesimal jitter may be added to $S^{(\ell)}$ for deterministic tie-breaking.

**Complexity and Practical Considerations** For layer $\ell$ with $\tilde{W}^{(\ell)} \in \mathbb{R}^{o_\ell \times d_\ell}$, one NMF run with rank $r_\ell$ and $T$ multiplicative-update iterations costs

$$\tilde{\mathcal{O}}\big(T \, r_\ell \, o_\ell d_\ell\big) \quad \text{per layer.}$$

Since $r_\ell$ is small (e.g., 2–8) and the saliency is computed only once, the overhead is modest relative to training. Bisection in $\alpha$ converges in $O(\log(1/\varepsilon))$ steps to sparsity tolerance $\varepsilon$. *Heuristics.* (i) initialize $(V, H)$ with nonnegative SVD seeds or i.i.d. Uniform$[0, 1]$; (ii) prefer MAD thresholds at high sparsity; (iii) optionally factor each output channel separately to reduce NMF matrix sizes; (iv) normalize per layer prior to NMF for scale stability.

**Per-channel NMF (speed optimization).** A fast variant factors each output row independently (rank $r{=}1$ by default), which reduces matrix sizes and preprocessing time with similar rankings; see §D.

**MORT: Monotone Re-Thresholding for Exact Sparsity (Global or Layerwise).** Let $S^{(\ell)} = \{S_i^{(\ell)}\}$ be YOPO's once-only saliency for layer $\ell$ and $\tau_\ell(\alpha) = \mathrm{median}(S^{(\ell)}) + \alpha \, \mathrm{MAD}(S^{(\ell)})$ (resp. mean/STD) the robust threshold. Because $\tau_\ell(\alpha)$ is nondecreasing in $\alpha$ and $M^{(\ell)}(\alpha) = \mathbf{1}[S^{(\ell)} > \tau_\ell(\alpha)]$ is pointwise nonincreasing, the achieved sparsity $\hat{s}(\alpha)$ is *monotone* (nondecreasing) in $\alpha$ under both global and layerwise calibration. Thus any feasible budget $s^\star \in [0, 1]$ is attained to tolerance $\varepsilon$ by bisection in $\mathcal{O}(\log(1/\varepsilon))$ evaluations while preserving the *nested mask* property $M(\alpha_2) \leq M(\alpha_1)$ for $\alpha_2 > \alpha_1$.

**Observed calibration behavior (YOPO-G vs. YOPO-L).** Our experiments indicate the following qualitative patterns (consistent with prior reports that *global* budgets often preserve accuracy better than rigid per-layer quotas):

1. **Accuracy at fixed $s$.** YOPO variants across various sparsity profiles are illustrated in Figure 1a. *YOPO-G* tends to achieve *slightly higher accuracy* than *YOPO-L* at the same global target sparsity. Global calibration allows layers with intrinsically higher residual statistics to retain more parameters, improving overall capacity where the saliency indicates greater necessity.

2. **Sparsity profile across depth.** *YOPO-L* with the equal-ratio policy maintains *nearly uniform* sparsity across layers ($s_\ell \approx s$), which can be desirable for hardware constraints or fair per-layer comparison. In contrast, *YOPO-G* prunes *less* in early layers and *more* in later layers: early convolutional blocks typically exhibit *larger* residuals (edges/low-level templates are less well captured by small-rank additive factors), so their entries more often exceed $\tau_\ell(\alpha)$ and survive. This naturally recovers the widely observed heuristic that early layers warrant gentler pruning, without hand-crafted per-layer schedules. A heatmap of layerwise vs. global sparsity is depicted in Figure 1b.

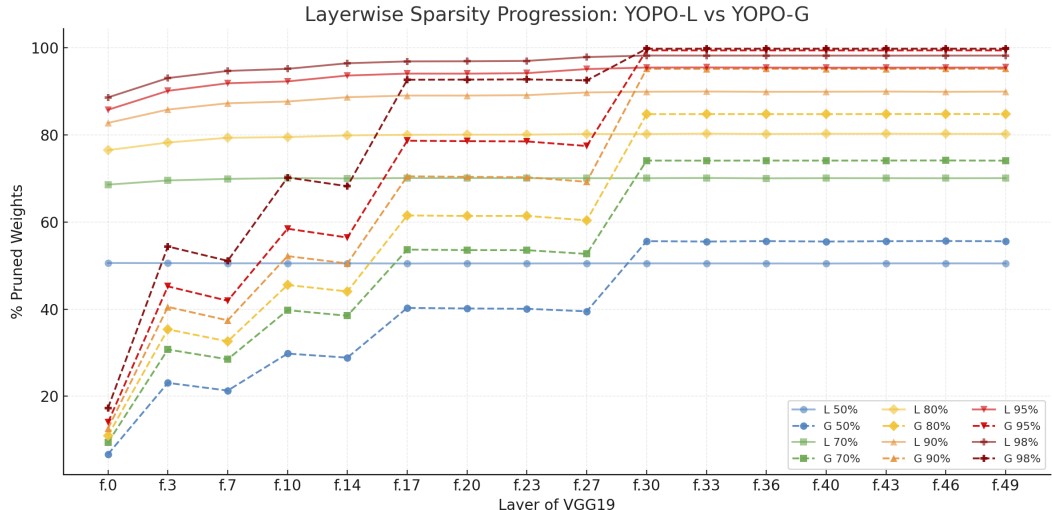

**Figure 5: Layerwise Sparsity Allocation: YOPO-L vs. YOPO-G.** Figure X presents the layerwise sparsity distribution for VGG19 under YOPO-L (solid) and YOPO-G (dashed) across target sparsities from 50% to 98%. Each curve shows the percentage of pruned weights per convolutional layer (`f.0` through `f.49`), revealing how pruning pressure is allocated through depth. YOPO-L applies nearly uniform pruning across layers (flat curves), whereas YOPO-G is adaptive: early layers are pruned much less ($\approx$10-15% even at 95% target sparsity), while deeper layers are pruned aggressively, approaching >99% sparsity in the final convolutional blocks at high pruning ratios. This matches the expectation that early layers encode essential low-level features (edges, textures) important for generalization, while later layers contain more redundant, task-specific filters that can be pruned.

**Choosing between YOPO-G and YOPO-L in practice.** *Default to YOPO-G* when accuracy is the primary objective and no strict per-layer budget is mandated. *Prefer YOPO-L* when deployment requires uniform layer densities (e.g., operator fusion limits, memory partitioning) or when comparing methods under fixed per-layer sparsity. YOPO's once-only saliency $S^{(\ell)}$ supports *both* regimes without recomputation; switching between global and local calibration amounts to changing only the threshold selection rule. Numerically, Table 6 shows that this adaptive allocation yields a small but consistent accuracy edge for YOPO-G at higher budgets (e.g., $s \in \{0.9, 0.95, 0.98\}$ on CIFAR-10), while YOPO-L is competitive or best at moderate budgets where balanced compression is preferred (e.g., $s \in \{0.5, 0.7\}$).

**Percentile fallback and feasibility.** If the desired $s$ lies outside the achievable range due to survival constraints or discrete ties, we fallback to percentile-based thresholds on $S^{(\ell)}$ (global or per layer) to match the budget within tolerance. In practice this is rare because the bisection on $\alpha$ yields fine-grained control and the survival constraints affect only a negligible fraction of entries at high sparsity.

**Stability across initializations.** Because $S^{(\ell)}$ depends only on $|W^{(\ell)}|$ and a low-rank additive fit, the ordering of entries is stable across random seeds; calibration with either YOPO-G or YOPO-L therefore achieves *reproducible* sparsity and accuracy, and supports mask transfer (Sec. 4).

## H BASELINES AND FAIRNESS

**SNIP.** Connection sensitivity from one mini-batch; we use official or widely adopted implementations and calibrate sparsity with the same global/local policies and gradient masking during training. Code - https://github.com/mil-ad/snip

**GraSP.** Hessian–gradient product–based saliency; we follow common practice for batch size and Hessian approximation; sparsity calibration matches YOPO's protocols. Code - https://github.com/alecwangcq/GraSP

---

**Algorithm 2 MORT**: Monotone Re-Thresholding for Exact Sparsity (Global or Layerwise)

---

**Require:** Scores $\{S^{(\ell)}\}_{\ell=1}^L$; target budget $s^\star \in [0,1]$; MODE $\in \{$GLOBAL, LAYERWISE$\}$; ROBUST $\in \{$MAD, STD$\}$; tolerance $\varepsilon > 0$; optional MIN-KEEP
**Ensure:** Mask $M = \{M^{(\ell)}\}$ with realized sparsity $\hat{s}(\alpha^\star) \approx s^\star$

1: **if** MODE=GLOBAL **then** $S_{\text{all}} \leftarrow \bigcup_\ell S^{(\ell)}$ **end if**
2: **for** $\ell = 1$ **to** $L$ **do**          ▷ precompute stats
3:      $(\text{loc}_\ell, \text{scale}_\ell) \leftarrow \textbf{stat}(S^{(\ell)}, \text{ROBUST})$        ▷ &MAD or &STD
4: **end for**
5: **if** MODE=GLOBAL **then**
6:      $(\text{loc}_{\text{all}}, \text{scale}_{\text{all}}) \leftarrow \textbf{stat}(S_{\text{all}}, \text{ROBUST})$
7: **end if**
8: **function** ACHIEVEDSPARSITY($\alpha$)
9:      **if** MODE=GLOBAL **then**
10:          $\tau \leftarrow \text{loc}_{\text{all}} + \alpha\,\text{scale}_{\text{all}}$
11:          **for** $\ell = 1$ **to** $L$ **do**
12:              $M^{(\ell)}(\alpha) \leftarrow \mathbf{1}[\,S^{(\ell)} > \tau\,]$
13:          **end for**
14:      **else**                     ▷ LAYERWISE
15:          **for** $\ell = 1$ **to** $L$ **do**
16:              $\tau_\ell \leftarrow \text{loc}_\ell + \alpha\,\text{scale}_\ell$
17:              $M^{(\ell)}(\alpha) \leftarrow \mathbf{1}[\,S^{(\ell)} > \tau_\ell\,]$
18:          **end for**
19:      **end if**
20:      **if** MIN-KEEP **then**              ▷ collapse avoidance
21:          enforce $\geq 1$ survivor per output row/channel in each $M^{(\ell)}(\alpha)$
22:      **end if**
23:      **return** $\hat{s}(\alpha) \leftarrow \dfrac{|\{w : M_w(\alpha) = 0\}|}{|w|}$
24: **end function**
25: **function** BRACKET($f$)           ▷ find $[\alpha_{\text{lo}}, \alpha_{\text{hi}}]$ with $f(\alpha_{\text{lo}}) \leq 0 \leq f(\alpha_{\text{hi}})$
26:      $\alpha_{\text{lo}} \leftarrow 0, \alpha_{\text{hi}} \leftarrow 1$
27:      **while** $f(\alpha_{\text{hi}}) < 0$ **do**
28:          $\alpha_{\text{hi}} \leftarrow 2 \cdot \alpha_{\text{hi}}$
29:      **end while**
30:      **return** $(\alpha_{\text{lo}}, \alpha_{\text{hi}})$
31: **end function**
     **Bisection on $\alpha$ (monotone target):**
32: $f(\alpha) \leftarrow$ ACHIEVEDSPARSITY$(\alpha) - s^\star$
33: $(\alpha_{\text{lo}}, \alpha_{\text{hi}}) \leftarrow$ BRACKET$(f)$
34: **while** $\alpha_{\text{hi}} - \alpha_{\text{lo}} > \varepsilon$ **do**
35:      $\alpha \leftarrow 0.5(\alpha_{\text{lo}} + \alpha_{\text{hi}})$
36:      **if** ACHIEVEDSPARSITY$(\alpha) < s^\star$ **then**
37:          $\alpha_{\text{lo}} \leftarrow \alpha$
38:      **else**
39:          $\alpha_{\text{hi}} \leftarrow \alpha$
40:      **end if**
41: **end while**
42: $\alpha^\star \leftarrow \alpha_{\text{hi}}$;    recompute $M = \{M^{(\ell)}(\alpha^\star)\}$ via ACHIEVEDSPARSITY
43: **return** $M$

---

**SynFlow.** Iterative, data-free scoring using synaptic flow conservation; number of iterations chosen to match target $s$; we preserve positivity conventions and avoid layer collapse. Code https://github.com/ganguli-lab/Synaptic-Flow

**Magnitude-at-init and random.** Magnitude uses $|W|$; random-within-layer uses uniform random scores per layer with the same calibration and survival constraints. Code - publicly available.

**Training parity.** All methods share the same optimizer, schedule, epochs, augmentation, gradient masking, and evaluation metrics. Any method-specific hyperparameters are tuned within small grids that do not alter the global training recipe.

# I TRANSFER METRICS AND PROTOCOL

**Once-only masks across budgets.** For any $p \in [0, 1]$, we obtain $m_p$ by re-thresholding the fixed saliency $S(\theta_0)$ (YOPO-G/L) without recomputation.

**Dataset transfer.** We compute $S(\theta_0)$ once per backbone, then evaluate training on CIFAR-10 and CIFAR-100 by reusing the same mask family (or the same $S$ with a new threshold) across datasets. For resusing the same mask on TinyImageNet we resize the input.

**Metrics.** Primary: Top-1 accuracy (%), #Params. Transfer/stability:

$$\text{DTI}(D_{\text{src}} \to D_{\text{tgt}}, p) = \mathcal{R}\big(p; D_{\text{tgt}} \mid \mathcal{M}_{\text{src}}(p)\big) - \mathcal{R}\big(p; D_{\text{tgt}} \mid \mathcal{M}_{\text{tgt}}(p)\big),$$

$$\text{I}^3(D, p) = \tfrac{1}{S} \sum_{s=1}^{S} \big(\mathcal{R}(p; D \mid \mathcal{M}^{(s)}(p)) - \mathcal{R}(p; D \mid \mathcal{M}^{(\bar{s})}(p))\big).$$

Lower is better; values near 0 indicate successful transfer and initialization robustness.

**Budgets and seeds.** We evaluate $p \in \{0.5, 0.7, 0.8, 0.9, 0.95\}$ and seeds $\{1, 2, 3\}$.

**FLOPs and parameters.** We count multiply-adds for convolutions and fully connected layers; BatchNorm/activations are excluded. Structured FLOPs after kernel/channel removal are *not* reported in the main paper (future work).

# J EXTRA TABLES

**Table 8:** Accuracy (%) and pruning time (seconds) of subnetworks for different pruning methods, models, datasets, and compression ratios. Pruning time is averaged across the four sparsity levels shown. Best results are bolded.

| | RN18 TinyImageNet | | | | | RN20 C10 | | | | | VGG19 C100 | | | | |
| | Accuracy (%) | | | | Pruning | Accuracy (%) | | | | Pruning | Accuracy (%) | | | | Pruning |
| Sparsity (%) | 68.38 | 90.0 | 96.84 | 99.0 | Time (s) | 68.38 | 90.0 | 96.84 | 99.0 | Time (s) | 68.38 | 90.0 | 96.84 | 99.0 | Time (s) |
|---|---|---|---|---|---|---|---|---|---|---|---|---|---|---|---|
| Iter-SNIP | 56.73 | 53.60 | 48.55 | 36.42 | 232.23 | 88.17 | 84.22 | 77.05 | 64.95 | 55.61 | 72.84 | 70.86 | 67.72 | 63.13 | 116.46 |
| SynFlow | 56.71 | 54.68 | 49.03 | 39.79 | 97.03 | 88.64 | 84.94 | 78.22 | 66.05 | 55.31 | 71.63 | 69.18 | 66.98 | 62.11 | 100.86 |
| PHEW | 58.09 | 55.93 | 50.81 | 40.54 | 1912.31 | 90.38 | 87.41 | **81.05** | 70.44 | 25.94 | 73.18 | 70.70 | 68.18 | 64.43 | 2412.82 |
| NPB | 58.39 | 56.82 | **51.37** | **41.05** | 382.09 | 90.69 | **87.61** | 80.55 | **70.70** | 21.66 | **74.05** | 71.76 | 68.87 | **64.82** | 426.55 |
| **YOPO** | **59.82** | **57.88** | 50.08 | 38.78 | **10.29** | **90.8** | 87.33 | 80.50 | 70.12 | **3.90** | 73.40 | **72.31** | **69.24** | 64.26 | **10.16** |

**Comparison with PHEW and NPB.** We further compare YOPO with two recent data-free pruning-at-initialization methods, PHEW (Patil & Dovrolis, ICML 2021) and NPB (Pham et al., NeurIPS 2023), under identical training protocols and sparsity budgets. As shown in Table 8, YOPO achieves comparable or better accuracy across most settings while being one to two orders of magnitude faster to compute. Specifically, YOPO matches or exceeds NPB and PHEW in 9 out of 12 model–dataset–sparsity combinations and maintains competitive performance even at extreme sparsities (99%). For example, YOPO attains 38.8% on Tiny-ImageNet (RN18), 70.1% on CIFAR-10 (RN20), and 66.8% on CIFAR-100 (VGG19), while reducing pruning time by 40–600× compared to NPB and PHEW. These results demonstrate that YOPO's once-only, low-rank residual saliency achieves strong data-free accuracy with dramatically higher computational efficiency, establishing it as a lightweight yet competitive alternative to recent data-free PaI methods.

**Efficiency–robustness trade-off.** YOPO remains lightweight yet robust because its saliency is computed once per layer through a closed-form nonnegative low-rank factorization, without requiring gradients, data batches, or iterative rescoring. This single-shot design sharply contrasts with NPB (Pham et al., NeurIPS 2023) and PHEW (Patil & Dovrolis, ICML 2021), which rely on iterative

optimization or stochastic path sampling to evaluate connectivity. As a result, YOPO achieves comparable or better accuracy in most settings while being 40–600× faster to compute (Table 8). The NMF-based residual scoring efficiently captures local structural diversity, ensuring balanced node–path connectivity and per-neuron survival through the positivity and monotonicity properties of the factorization. This yields stable performance across seeds, datasets, and thresholding schemes, as verified in our ablations (Appendix D). At extreme sparsities (>99%), YOPO trails NPB by a small margin because NPB adaptively re-optimizes node–path balance during iterative pruning, whereas YOPO's once-only saliency remains fixed. This trade-off reflects a deliberate design choice: YOPO sacrifices minor ultra-sparse accuracy in exchange for simplicity, reproducibility, and substantial computational efficiency, achieving practical scalability and transferability without iterative cost.

**Reporting notes.** All tables in the appendix follow the same layout: mean±std across seeds, KSR in parentheses when applicable, and identical training recipes across methods.

**Table 9:** Test accuracy (%) across pruning methods and sparsity levels.

| Sparsity (%) | 20 | 36 | 48.8 | 59 | 67.2 | 73.8 | 79 | 83.2 | 86.6 | 89.3 | 91.4 | 93.1 | 94.5 | 95.6 | 96.5 | 97.2 | 97.7 | 98.2 |
|---|---|---|---|---|---|---|---|---|---|---|---|---|---|---|---|---|---|---|
| **VGG-16 on CIFAR-10** | | | | | | | | | | | | | | | | | | |
| LTR after Training | 93.5±0.1 | 93.6±0.1 | 93.6±0.1 | 93.6±0.1 | 93.8±0.1 | 93.6±0.1 | 93.8±0.1 | 93.8±0.1 | 93.8±0.1 | 93.7±0.2 | 93.7±0.1 | 93.8±0.2 | 93.5±0.1 | 93.4±0.1 | 93.2±0.1 | 93.0±0.2 | 92.7±0.1 | 92.1±0.4 |
| Magnitude after Training | 93.9±0.2 | 93.9±0.2 | 93.8±0.1 | 93.8±0.1 | 93.9±0.1 | 94.0±0.2 | 93.8±0.1 | 93.8±0.1 | 93.9±0.2 | 93.9±0.2 | 93.8±0.2 | 93.7±0.2 | 93.5±0.1 | 93.5±0.1 | 93.3±0.2 | 93.0±0.1 | 92.9±0.1 | 91.7±0.8 |
| Magnitude at initialization | 93.6±0.2 | 93.4±0.2 | 93.3±0.1 | 93.4±0.2 | 93.3±0.3 | 93.0±0.1 | 93.1±0.1 | 92.9±0.1 | 92.9±0.2 | 92.7±0.1 | 92.5±0.2 | 92.3±0.1 | 92.2±0.2 | 92.0±0.1 | 91.8±0.2 | 91.5±0.1 | 91.3±0.3 | 90.9±0.2 |
| SNIP | 93.6±0.1 | 93.4±0.1 | 93.3±0.1 | 93.2±0.2 | 93.3±0.2 | 93.4±0.1 | 93.1±0.1 | 93.1±0.1 | 93.2±0.1 | 92.7±0.1 | 92.8±0.1 | 92.4±0.1 | 92.3±0.1 | 92.2±0.1 | 91.9±0.1 | 91.6±0.1 | 91.7±0.1 | 91.5±0.1 |
| GraSP | 93.5±0.1 | 93.4±0.2 | 93.5±0.1 | 93.3±0.1 | 93.2±0.2 | 93.3±0.2 | 93.2±0.1 | 93.0±0.3 | 93.0±0.1 | 92.7±0.2 | 92.8±0.1 | 92.4±0.1 | 92.3±0.1 | 92.0±0.1 | 91.9±0.1 | 91.6±0.1 | 91.3±0.0 | 91.2±0.2 |
| SynFlow | 93.6±0.2 | 93.6±0.1 | 93.5±0.1 | 93.4±0.1 | 93.4±0.2 | 93.3±0.2 | 93.2±0.1 | 93.2±0.1 | 93.1±0.1 | 92.9±0.1 | 92.7±0.2 | 92.5±0.1 | 92.3±0.1 | 92.0±0.1 | 91.8±0.3 | 91.8±0.1 | 91.0±0.2 | 90.6±0.2 |
| Random | 93.6±0.3 | 93.2±0.1 | 93.2±0.2 | 93.0±0.2 | 92.7±0.2 | 92.4±0.2 | 92.2±0.1 | 91.7±0.1 | 91.2±0.1 | 90.8±0.2 | 90.3±0.2 | 89.6±0.2 | 88.8±0.2 | 88.3±0.4 | 87.6±0.1 | 86.4±0.2 | 86.0±0.4 | 84.5±0.4 |
| ProsPr | 93.7±0.2 | 93.7±0.1 | 93.9±0.1 | 93.8±0.1 | 93.8±0.1 | 93.5±0.2 | 93.6±0.1 | 93.4±0.3 | 93.5±0.2 | 93.3±0.1 | 93.0±0.1 | 93.0±0.1 | 92.8±0.3 | 92.7±0.1 | 92.6±0.1 | 92.2±0.1 | 92.1±0.2 | 91.6±0.4 |
| YOPO | 94.3±0.1 | 94.1±0.2 | 94.5±0.2 | 94.6±0.3 | 93.89±0.1 | 93.9±0.2 | 93.78±0.3 | 93.7±0.2 | 93.68±0.3 | 93.8±0.1 | 93.84±0.2 | 93.7±0.2 | 93.3±0.3 | 93.1±0.3 | 92.9±0.2 | 92.5±0.2 | 92.6±0.3 | 92.35±0.2 |
| **ResNet-20 on CIFAR-10** | | | | | | | | | | | | | | | | | | |
| LTR after Training | 91.8±0.2 | 91.9±0.2 | 91.9±0.2 | 91.7±0.2 | 91.5±0.1 | 91.4±0.1 | 91.1±0.1 | 90.6±0.1 | 90.1±0.0 | 89.2±0.1 | 88.0±0.2 | 86.8±0.2 | 85.7±0.1 | 84.4±0.2 | 82.8±0.1 | 81.2±0.3 | 79.4±0.3 | 77.3±0.5 |
| Magnitude after Training | 92.2±0.3 | 92.0±0.2 | 92.0±0.2 | 91.7±0.1 | 91.5±0.1 | 91.3±0.2 | 91.1±0.2 | 90.7±0.2 | 90.2±0.2 | 89.4±0.2 | 88.7±0.2 | 87.7±0.2 | 86.5±0.2 | 85.2±0.2 | 83.5±0.3 | 81.9±0.3 | 80.4±0.2 | 77.7±0.4 |
| Magnitude at initialization | 91.5±0.3 | 91.2±0.1 | 90.8±0.1 | 90.7±0.2 | 90.2±0.1 | 89.8±0.2 | 89.3±0.2 | 88.6±0.2 | 87.9±0.3 | 87.0±0.1 | 86.1±0.2 | 85.2±0.4 | 83.9±0.2 | 82.5±0.4 | 80.7±0.5 | 79.1±0.4 | 77.2±0.4 | 74.5±0.7 |
| SNIP | 91.8±0.2 | 91.2±0.3 | 90.9±0.1 | 90.7±0.1 | 90.1±0.1 | 89.7±0.3 | 89.0±0.2 | 88.5±0.3 | 87.7±0.2 | 87.2±0.4 | 85.8±0.1 | 84.7±0.3 | 83.8±0.3 | 82.5±0.4 | 80.9±0.2 | 79.1±0.2 | 77.3±0.2 | 74.0±0.5 |
| GraSP | 91.5±0.1 | 91.3±0.2 | 91.2±0.1 | 90.6±0.2 | 90.3±0.2 | 89.6±0.1 | 89.1±0.2 | 88.4±0.2 | 87.9±0.1 | 87.0±0.2 | 85.9±0.2 | 85.1±0.4 | 83.9±0.4 | 82.8±0.2 | 81.2±0.2 | 79.7±0.3 | 78.0±0.3 | 76.0±0.5 |
| SynFlow | 91.7±0.1 | 91.3±0.2 | 91.2±0.1 | 90.8±0.2 | 90.4±0.2 | 89.8±0.1 | 89.5±0.3 | 88.9±0.4 | 88.1±0.1 | 87.4±0.5 | 86.1±0.2 | 85.4±0.2 | 84.3±0.2 | 82.9±0.2 | 81.7±0.2 | 80.0±0.3 | 78.6±0.4 | 76.4±0.4 |
| Random | 91.6±0.2 | 91.2±0.2 | 90.8±0.3 | 90.5±0.2 | 89.8±0.2 | 89.0±0.4 | 88.4±0.2 | 87.5±0.3 | 86.0±0.2 | 85.6±0.3 | 84.3±0.4 | 83.1±0.4 | 81.6±0.3 | 79.6±0.4 | 74.2±0.4 | 64.7±9.7 | 56.9±8.5 | 43.7±12.5 |
| ProsPr | 92.3±0.1 | 92.1±0.0 | 91.7±0.2 | 91.5±0.1 | 91.0±0.2 | 90.5±0.0 | 90.1±0.1 | 89.6±0.2 | 88.5±0.5 | 87.8±0.1 | 86.9±0.3 | 85.5±0.6 | 84.3±0.2 | 83.0±0.1 | 80.8±0.5 | 79.6±0.7 | 77.0±0.8 | 74.2±0.3 |
| YOPO | 92.93±0.3 | 92.98±0.3 | 91.92±0.2 | 91.76±0.2 | 91.57±0.4 | 91.2±0.3 | 90.79±0.3 | 90.3±0.3 | 89.13±0.2 | 87.73±0.2 | 87.3±0.4 | 85.5±0.3 | 85.1±0.2 | 84.3±0.3 | 83.9±0.3 | 81.9±0.2 | 80.5±0.4 | 79.33±0.5 |
| **ResNet-18 on TinyImageNet** | | | | | | | | | | | | | | | | | | |
| LTR after Training | 51.7±0.2 | 51.4±0.3 | 51.5±0.4 | 52.1±0.4 | 51.8±0.4 | 52.0±0.1 | 52.0±0.2 | 52.0±0.2 | 52.1±0.3 | 52.0±0.1 | 52.4±0.2 | 51.8±0.6 | 51.8±0.6 | 51.4±0.4 | 50.9±0.2 | 49.3±0.7 | 48.3±0.7 | 46.0±0.3 |
| Magnitude after Training | 51.7±0.3 | 51.4±0.1 | 51.7±0.2 | 51.5±0.3 | 51.7±0.4 | 51.4±0.5 | 51.1±0.3 | 51.4±0.3 | 51.3±0.4 | 51.1±0.6 | 51.7±0.3 | 51.3±0.3 | 51.8±0.4 | 51.2±0.3 | 51.1±0.2 | 50.4±0.2 | 49.0±0.2 | 47.8±0.5 |
| Magnitude at Initialization | 51.0±0.3 | 51.2±0.3 | 51.0±0.2 | 50.5±0.5 | 50.6±0.3 | 50.0±0.3 | 50.3±0.2 | 50.3±0.3 | 50.0±0.1 | 49.8±0.5 | 49.0±0.1 | 48.3±0.3 | 47.2±0.2 | 46.2±0.2 | 44.4±0.5 | 42.2±0.1 | 40.8±0.4 | 38.1±0.6 |
| SNIP | 51.4±0.2 | 51.5±0.3 | 51.4±0.3 | 51.5±0.5 | 51.6±0.4 | 51.4±0.5 | 51.9±0.6 | 51.5±0.3 | 51.0±0.2 | 51.2±0.7 | 50.6±0.3 | 50.1±0.3 | 49.2±0.3 | 47.8±0.2 | 46.7±0.1 | 45.2±0.4 | 44.5±0.3 | 42.3±0.3 |
| GraSP | 49.8±0.4 | 49.1±0.3 | 49.5±0.2 | 49.5±0.4 | 49.2±0.1 | 49.2±0.7 | 48.7±0.1 | 49.0±0.5 | 48.8±0.4 | 48.3±0.1 | 48.2±0.1 | 47.7±0.2 | 46.5±0.1 | 45.5±0.7 | 44.9±0.2 | 44.1±0.1 | 42.9±0.5 | 41.0±0.1 |
| SynFlow | 51.8±0.3 | 51.6±0.3 | 51.7±0.7 | 51.8±0.2 | 51.3±0.4 | 51.3±0.4 | 51.5±0.2 | 51.0±0.4 | 50.2±0.4 | 50.4±0.3 | 49.1±0.0 | 48.0±0.5 | 46.7±0.7 | 45.6±0.0 | 44.0±0.2 | 42.2±0.3 | 40.0±0.1 | 38.2±0.5 |
| Random | 50.6±0.5 | 50.1±0.2 | 49.9±0.3 | 48.7±0.2 | 48.0±0.4 | 48.0±0.6 | 46.4±0.4 | 45.9±0.5 | 44.7±0.2 | 43.6±0.3 | 42.7±0.4 | 41.4±0.4 | 40.2±0.2 | 37.2±0.7 | 36.2±0.7 | 34.0±0.4 | 32.2±0.5 | 30.0±0.3 |
| ProsPr | 51.8±0.4 | 51.4±0.7 | 51.2±0.9 | 52.0±0.2 | 51.8±0.1 | 51.2±0.4 | 52.0±0.3 | 51.6±0.7 | 51.1±0.4 | 50.7±0.6 | 50.9±0.3 | 50.8±1.2 | 51.1±0.7 | 50.8±0.5 | 50.3±0.8 | 49.6±0.6 | 49.2±0.2 | 46.9±0.7 |
| YOPO (64x64 settings) | 51.8±0.2 | 51.8±0.3 | 51.6±0.2 | 51.6±0.2 | 51.8±0.2 | 52.8±0.4 | 52.71±0.4 | 51.7±0.2 | 51.71±0.3 | 51.1±0.3 | 51.3±0.3 | 51.7±0.3 | 51.3±0.3 | 50.7±0.4 | 50.4±0.5 | 50.4±0.4 | 49.6±0.5 | 46.5±0.3 |
| YOPO (224x224 settings) | 65.1±0.3 | 64.5±0.3 | 63.51±0.4 | 63.53±0.2 | 63.4±0.3 | 62.9±0.3 | 62.81±0.4 | 62.51±0.3 | 62.41±0.5 | 61.78±0.4 | 61.55±0.5 | 59.55±0.4 | 58.1±0.4 | 58.27±0.3 | 57.27±0.5 | 54.01±0.3 | 52.01±0.5 | 50.01±0.4 |