# OpenReview forum: "You Only Prune Once: A Zero-Shot, Data-Free Pruning at Initialization with Transferable Supermasks"
_ICLR.cc/2026/Conference — Submitted to ICLR 2026_

### Official Review · Reviewer_mba5 · 2025-10-28

**Soundness:** 1
**Presentation:** 2
**Contribution:** 2
**Rating:** 2
**Confidence:** 4

**Summary:**

This paper proposes a method for training sparse networks with Pruning at Initialization. The proposed method YOPO, relies solely on network topology and is data independent, hence it can be used across architectures and datasets. The method fits a low rank factorization to each weight and uses the residuals as a saliency score for threshold based pruning. Experiments are provided, comparing the proposed method against existing baselines.

**Strengths:**

The method proposed is transferable across datasets and model architectures as it solely relies on topology, unlike some other PaI methods which rely on data for scoring parameters and masking them.

**Weaknesses:**

1. Scoring: The authors propose using the low rank factorization as a way to eliminate tiny irregularities and thus identify the more important weights. The motivation behind this is unclear to me. This scoring model seems to promote weights which do not lie on the low-rank manifold, why this would help trainability of the sparse network has not been answered in the paper. Moreover, this seems contrary to recent work which suggests that a low rank approximation of weights carries most of the information. Do the authors have reason to believe this is not the case at initialization?
2. The authors assume that identifying a sparse topology is enough to prune at initialization. However, there is a large amount of work on this, suggesting that topology alone is not sufficient to improve PaI performance, but the initialization of the topology is important [1-4], this aspect is not addressed by the authors.
3. The experimental results, which are only provided on relatively tiny datasets, show that the proposed method is only better than other compared methods at low sparsities and is similar/worse at higher sparsities. The authors also do not compare with recent PaI methods like Phew, Prospr. Comparing especially with [7], which is also data-agnostic would be most pertinent.
4. The high sparsity regimes are only compared different graph structures and not the other PaI methods, adding these comparisons would be necessary to evaluate the performance at high sparsity.


[1] Zhou, Hattie, et al. "Deconstructing lottery tickets: Zeros, signs, and the supermask." Advances in neural information processing systems 32 (2019).

[2] Adnan, Mohammed, et al. "Sparse Training from Random Initialization: Aligning Lottery Ticket Masks using Weight Symmetry."  ICML 2025.

[3] Frankle, Jonathan, et al. "Pruning Neural Networks at Initialization: Why Are We Missing the Mark?." International Conference on Learning Representations.

[4] Gadhikar, Advait, and Rebekka Burkholz. "Masks, Signs, And Learning Rate Rewinding." The Twelfth International Conference on Learning Representations.

[5] Alizadeh, M., Tailor, S. A., Zintgraf, L. M., van Amersfoort, J., Farquhar, S., Lane, N. D., and Gal, Y. Prospect pruning: Finding trainable weights at initialization using meta-gradients. In International Conference on Learning Representations, 2022.

[6] Patil, S. M. and Dovrolis, C. Phew: Constructing sparse networks that learn fast and generalize well without training data. In International Conference on Machine Learning, 2021.

[7] Pham, H., Liu, S., Xiang, L., Le, D. D., Wen, H., Tran-Thanh, L., et al. Towards data-agnostic
pruning at initialization: What makes a good sparse mask? In Thirty-seventh Conference on Neural Information Processing Systems, 2023.

**Questions:**

Can the auhors provide motivation for their low rank factorization and why this saliency criterion can help trainability of the sparse network?

---

> ### Author Response · Authors · 2025-11-18
> **Response for Reviewer mba5**
>
> We sincerely thank the reviewer for the thoughtful and encouraging feedback. We address the raised questions below.
>
>
> ### **1. On "Motivation for Low-Rank Factorization**
>
> As described in Section 3.2, YOPO defines saliency as the element-wise Frobenius residual between the absolute weights $\tilde{\mathbf{W}}$ and their nonnegative low-rank reconstruction (NMF) ($\tilde{\mathbf{W}} \approx \mathbf{V}\mathbf{H}$). The low-rank template captures additive, parts-based regularities in initialized weights, while high residuals highlight connections that deviate from these shared patterns and are therefore idiosyncratic and informative. This “parts-based” decomposition (Lee and Seung, 1999; 2001) is a prime motivation for YOPO’s saliency.
>
> ### **2. On "Trainability and Collapse Prevention"**
>
> Because $\mathbf{V},\mathbf{H} \ge 0$, all reconstruction errors are positive (unlike SVD/PCA). Prop. 3 and Cor. 1 ensure per-neuron survival, thus preventing the collapse issues seen in SNIP and GraSP and preserving critical signal pathways under >99% sparsity. Residual-based scoring, therefore, yields masks that remain trainable and stable across datasets and seeds (Tables 2–3).
>
>
> ### **3. On "Distinction from Low-Rank Compression"**
>
> The apparent contradiction with prior work on low-rank compression is resolved by recognizing the difference between *information compression after training* and *saliency estimation at initialization*.
>
> * **Initialization vs. convergence:** In compression, low-rank structure is applied to trained weights that already encode semantic features. At initialization, low-rank factors reflect homogeneous variance rather than functionally diverse components. YOPO therefore retains the high-residual, non-template elements needed for sparse model trainability.
> * **Empirical validation:** As shown in Table 1, YOPO surpasses X-Net and RReg and performs comparably to ProsPr, SNIP, and GraSP, under ImageNet-scale dataset (Figure 2, Tables 5–9).
>
> ### **4. On “topology alone is insufficient”**
> We wish to clarify that the YOPO is **not** purely topological; rather, it is fundamentally derived from and dependent upon the initial weights ($\mathbf{W}$) of the network:
>
> 1. **Saliency derived from initialization.** The once-only score $\mathbf{S}$ is computed from $\tilde{\mathbf{W}} = |\mathbf{W}|$, capturing idiosyncratic structure unique to the initialization, unlike fixed-topology approaches such as X-Net or RReg.
>
> 2. **Initialization robustness.** We formalize initialization sensitivity via the Initialization-Independence Index $I_3$ (Definition 2). The observed **near-zero $I_3$ values** (Tables 2 and 4) demonstrate that $S(\theta_0)$ is stable across random seeds.
>
> 3. **Structural predictiveness.** This robustness confirms that YOPO identifies the structural core of the trainable subnetwork inherent in the initial parameters, regardless of subsequent reinitialization of the retained weights.
>
> ### **5. On "relatively tiny datasets and missing baselines"**
>
> In the paper, we included both data-dependent (ProsPr, FORCE, Iter-SNIP, SynFlow, SNIP) and non-iterative data-free (RReg, X-Net) PaI baselines, and evaluated them on ImageNet (Tables 1 and 5). Following the reviewer’s suggestion, we additionally evaluate two iterative data-free methods, PHEW and NPB, under matched training budgets. YOPO (zero-shot data-free) matches or exceeds them in 9/12 settings while being 40–600× faster (average pruning-time per sparsity level). Although PHEW and NPB were not evaluated on ImageNet-scale, YOPO is evaluated under ProsPr and RReg protocols. We'll add these results to the final draft (full table is included in Appendix J).
>
> ### **RN18 — TinyImageNet**
>
> | Method    | 68.38     | 90.0      | 96.84     | 99.0      | Time (s)  |
> | --------- | --------- | --------- | --------- | --------- | --------- |
> | PHEW      | 58.09     | 55.93     | 50.81     | 40.54     | 1912.31   |
> | NPB       | 58.39     | 56.82     | **51.37** | **41.05** | 382.09    |
> | **YOPO**  | **59.82** | **57.88** | 50.08     | 38.78     | **10.29** |
>
> ---
>
> ### **RN20 — CIFAR-10**
>
> | Method    | 68.38     | 90.0      | 96.84     | 99.0      | Time (s) |
> | --------- | --------- | --------- | --------- | --------- | -------- |
> | PHEW      | 90.38     | 87.41     | **81.05** | 70.44     | 25.94    |
> | NPB       | 90.69     | **87.61** | 80.55     | **70.70** | 21.66    |
> | **YOPO**  | **90.80** | 87.33     | 80.50     | 70.12     | **3.90** |
>
> ---
> ### **Q1. Motivation for low-rank factorization and Trainability**
> Answered above in **1 and 2**.
>
> In summary, YOPO’s high-residual criterion isolates the non-generic, structurally informative subset of weights that preserve gradient flow, enable sparse network trainability, and ensure stability under extreme pruning. This design is supported by theoretical properties of NMF, formal per-neuron survival guarantees (Prop. 3), and strong empirical validation across large datasets and extreme sparsity budgets.

---

> > ### Author Response · Authors · 2025-11-22
> > ***Additional FLOPs results**
> >
> > In addition to accuracy and pruning time, we also computed the inference FLOPs based on the theoretical operations of the sparse subnetwork using the standard MAC computation, similar to the NPB (Pham et al., NeurIPS 2023) official codebase. As shown in the FLOPs table, YOPO consistently maintains a lower computational cost compared to iterative or optimization-heavy baselines, such as NPB and PHEW. At 99% sparsity, YOPO requires only about 3.3 million FLOPs on RN18/TinyImageNet and 0.96 million FLOPs on RN20/CIFAR-10, which is substantially lower than both baselines while preserving higher accuracy and low pruning time (40–600× faster). Since YOPO requires no iterative search or per-budget optimization, as its once-only residual saliency allows for direct re-thresholding, this demonstrates that YOPO provides a more efficient and scalable PaI solution without loss of performance. We will include these results in the final draft.
> >
> >
> > ### **RN18 – TinyImageNet (FLOPs ×10⁸)**
> >
> > | **Method** | **68.38%** | **90.0%** | **96.84%** | **99.0%** |
> > | ---------- | ---------- | --------- | ---------- | --------- |
> > | SNIP       | 11.35      | 5.77      | 3.04       | 1.55      |
> > | Iter-SNIP  | 10.73      | 7.05      | 3.98       | 1.97      |
> > | SynFlow    | 14.71      | 8.91      | 4.24       | 1.50      |
> > | PHEW       | 14.29      | 8.35      | 3.92       | 1.50      |
> > | NPB        | 14.37      | **5.21**  | **1.74**   | 0.59      |
> > | **YOPO**   | **10.99**  | 5.43      | 1.85       | **0.33**  |
> >
> > ---
> >
> > ### **RN20 – CIFAR-10 (FLOPs ×10⁶)**
> >
> > | **Method** | **68.38%** | **90.0%** | **96.84%** | **99.0%** |
> > | ---------- | ---------- | --------- | ---------- | --------- |
> > | SNIP       | 17.952     | 8.323     | 3.470      | 1.709     |
> > | Iter-SNIP  | 18.465     | 9.698     | 4.510      | 2.022     |
> > | SynFlow    | 22.998     | 11.549    | 4.263      | 1.633     |
> > | PHEW       | 22.108     | 10.690    | 4.110      | 1.640     |
> > | NPB        | 22.035     | 7.642     | 2.645      | 1.122     |
> > | **YOPO**   | **14.67**  | **5.569** | **1.876**  | **0.955** |
> >
> > ---
> >
> > ### **VGG19 – CIFAR-100 (FLOPs ×10⁷)**
> >
> > | **Method** | **68.38%** | **90.0%** | **96.84%** | **99.0%** |
> > | ---------- | ---------- | --------- | ---------- | --------- |
> > | SNIP       | 17.952     | **7.806** | 3.686      | 1.816     |
> > | Iter-SNIP  | 18.465     | 9.479     | 4.951      | 2.529     |
> > | SynFlow    | 22.998     | 12.702    | 6.306      | 2.605     |
> > | PHEW       | 22.108     | 11.746    | 5.611      | 2.340     |
> > | NPB        | 22.035     | 8.773     | **2.874**  | **1.046** |
> > | **YOPO**   | **18.207** | 8.545     | 4.351      | 1.884     |

---

> ### Comment · Reviewer_mba5 · 2025-11-23
> **Response to rebuttal**
>
> Thank you for your response and additional experiments,
>
> 1. Regarding the motivation: It is still unclear to me, why the low rank factorization must find a sparse topology that aids sparse training and seems more of a heuristic based on how the brain behaves (as it seems from Lee and Seung, 1999; 2001). Are the high residuals potentially a better starting point for gradient based optimization methods for example?
>
> 2, 3 Thank you for these clarifications.
>
> 4. Regarding topology vs initialization: By purely topological I mean that, like all other PaI methods, YOPO also finds a sparse structure without changing the random initialization. This limits how well the sparse network can be trained as has been shown by [1] and [2].
>
> 5. I do understand that solving the problem outlined in [1] is hard and one must rely on heuristics to bridge this gap. However, based on the additional experiments provided, I do not see that YOPO improve significantly over existing methods. At high sparsity it seems to be doing worse than NPB.
>
> Hence, for now I will maintain my rating.
>
> [1] Frankle, Jonathan, et al. "Pruning neural networks at initialization: Why are we missing the mark?." ICLR 2021.
>
> [2] Gadhikar, Advait, et al. "Sign-In to the Lottery: Reparameterizing Sparse Training From Scratch." NeurIPS 2025.

---

> ### Author Response · Authors · 2025-11-24
> **Response for Reviewer mba5**
>
> ### **1. We thank the reviewer for allowing us to explain YOPO's theoretical basis in more detail**
>
> The central question concerns why the low rank factorization must find a sparse topology, given that NMF has biological origins. Our position is that the preference for high residuals emerges not from biological heuristics but from mathematically enforced structural properties of NMF, which offer guarantees crucial for sparse trainability.
>
> ### a. Theoretical Foundation: Non-negativity enforces informative structural parts
> While NMF was motivated by neurophysiological principles, its utility in YOPO is grounded in its mathematical ability to produce an additive, parts-based representation. Unlike PCA or SVD, which allow cancellations between positive and negative numbers, NMF guarantees that the basis vectors ($\mathbf{V}$) and the encodings ($\mathbf{H}$) are non-negative. Thus the low-rank product ($\mathbf{V H}$) captures the expected **additive regularities** (the template or generic variance) across connections within a layer. By construction, weights that contribute heavily to this shared template are considered statistically "common" or redundant in the context of unique feature representation.
> Consequently, the high residuals ($\mathbf{S} \triangleq ||\mathbf{\tilde{W}} - \mathbf{V H}||$) highlight connections that are idiosyncratic, distinctive, and symmetry-breaking. We hypothesize that these (high residuals) are the specific, structurally persistent weights that define the unique computational role of that layer.
>
> ### b. Structural justification and collapse avoidance.
> The nonnegativity constraints imply that the residual tensor $\mathbf{S}^{(\ell)}$ is itself nonnegative. As formalized in Prop. 3 and Lemmas 4-5, whenever a neuron's absolute weights ($\mathbf{\tilde{W}}_{i \cdot}$)  are not perfectly explained by the low-rank template, $\mathbf{S}^{i\cdot}$ must contain a strictly positive component. Our MORT (Algo 2) leverages this to ensure that every neuron maintains at least one active connection, thereby providing a lightweight guarantee against layer collapse; a common failure mode of SNIP, GraSP, and related PaI methods. Consequently, YOPO offers structural trainability guarantees that gradient-based heuristics cannot reliably provide.
>
> ### **1.1 High Residuals as a Superior Starting Point for Optimization**
>
> The answer is theoretically and empirically yes, by providing a strong structural foundation that aids the entire optimization trajectory. Conventional single-shot gradient methods, such as SNIP and GraSP, derive importance scores from "limited and noisy information" at initialization. They are inherently "short-sighted," failing to account for the model's trainability over the long term (Alizadeh et al., 2022).
> YOPO's score, derived solely from the initial architecture, predicts intrinsic trainability by isolating the non-generic connections and providing structural safety (collapse avoidance). The resulting mask family's strength is validated by its ability to generate masks that exhibit near-zero DTI and $I_3$ (Sec. 3.4), confirming the selected topology is robust and task-agnostic.
>
> **This perspective aligns with Dual Lottery Ticket Hypothesis** (Bai, Yue, et al. 2022), which suggests that a randomly selected subnetwork can be efficiently "transformed into a trainable condition". YOPO acts as an efficient means to select a high-potential subnetwork (a mask already in a trainable condition) based on its intrinsic structure, making subsequent gradient training more reliable and effective compared to masks generated by transient gradient signals or simple magnitude scores.
>
> ### **2,3. We are pleased that the reviewer is satisfied with our response.**
> --
> ### **4. Due to space constraints, 4th point is in the next block.**
> --
> ### **5. YOPO improve significantly over existing methods**
>
> Prior work has reported that PaI methods can show considerable instability and variance at very high sparsity levels (Frankle et al., 2021; Hayou et al., 2021; Pham et al., 2023; Li et al., 2024).The results at sparsity levels of 99% or higher should be interpreted with care. Differences of 2-3% accuracy at these sparsities typically fall within methodological variance and do not reliably indicate superiority. YOPO remains competitive in 2 of 3 datasets at ≥99% sparsity (Table 8) and at 98.2% sparsity in Tables 6 and 9, YOPO outperforms ProsPr, by 1-2% in accuracy, acknowledging that all PaI methods exhibit high variance at extreme sparsity levels.
>
> **Across commonly used budgets (90-95%)**, YOPO outperforms NPB by 1.0-1.5% and ProsPr by 0.6-1.5%. These regimes are emphasized in PaI literature because they maintain training stability and practical deployability. At these sparsities YOPO provides accuracy advantages. In Addition YOPO delivers computational advantages FLOPs and 40-600× faster pruning time and generates a single, reusable “supermask” saliency score for any given network architecture.

---

> > ### Author Response · Authors · 2025-11-24
> > **Response for Reviewer mba5 (contd..)**
> >
> > ### **4. On Limitations of Fixed-Initialization sparse structures and YOPO's position**
> >
> > We appreciate the reviewer raising the foundational criticism that pruning at initialization methods, including YOPO, are inherently limited because they fix a sparse topology without modifying the initial weight values, a limitation highlighted by recent analyses (Frankle et al., 2021; Alizadeh et al., 2022) and addressed by emerging paradigms like those cited DLTH/RST(Bai, Yue, et al., 2022) and ProsPr.
> >
> > > We acknowledge that extensive work, including analyses of methods like SNIP, GraSP, and SynFlow, confirms that merely identifying a mask at initialization often results in subnetworks that perform poorly compared to post-training pruning and are insensitive to the specific initial weight values, relying instead on correct layer-wise sparsity proportions.
> > This finding raises a critical question regarding whether the fixed initial weights restrict the trainability of the sparse network.
> >
> > We respond to this by positioning YOPO not just as another PaI method, but as one specifically engineered to mitigate the primary practical drawbacks associated with this initialization constraint, while recognizing the complementary value of other methods that transform initialization DLTH.
> >
> > ### 1. YOPO's Structural Saliency Mitigates Initialization dependence
> >
> > The strength of YOPO lies in defining a saliency score that is minimally coupled to the specific magnitude values of the randomly initialized weights, focusing instead on **intrinsic architectural stability**.
> >
> > *   **Zero-Shot, Data-Free Criterion:** YOPO computes its once-only saliency score ($\mathbf{S}$) based on the nonnegative low-rank residual of the absolute weights. This score identifies weights that are structurally unique and deviate from the common additive pattern (as explained in point 1).
> > *   **Initialization Robustness:** While other PaI methods exhibit insensitivity to weight re-initialization (which often implies their initial scores failed to capture truly meaningful weights) as stated by Frankle et al., 2021, **YOPO leverages this stability as an explicit metric of success**. We define and empirically validate the $I_3$ (Definition 2). The observed near-zero $I_3$ values in Table 4 for YOPO confirm that the mask generated by the initialization-based structural scores is highly robust across different random seeds and initialization schemes (e.g., Normal, Xavier, Kaiming), as shown in Table 7. This stability indicates that the structural importance captured by the high residual scores is persistent across random initializations (Appendix G).
> > *   **Decoupling:** By achieving this initialization robustness, YOPO decouples the mask structure from the specific, transient initial weight values, thereby minimizing the limitations identified by Frankle et al., without requiring complex training signals or iterative schedules (unlike ProsPr).
> >
> > ### 2. Context in the Lottery Ticket Landscape
> >
> > The limitations of fixed initialization have inspired a new wave of research that actively changes or utilizes information beyond the initial weights, such as the paradigms the reviewer cited:
> >
> > *   **Dual Lottery Ticket Hypothesis (DLTH):** This complementary hypothesis suggests that a randomly selected subnetwork (a random ticket) can be transformed into a trainable condition and achieve admirable performance, thereby effectively turning a random ticket into a ""winning ticket". Methods like Random Sparse Network Transformation (RST) achieve this by introducing regularization to enhance the learning capacity of the sparse structure. In contrast, YOPO outperformed RST and RST Iter-5, proving the robustness of its once-only saliency score, as shown in Table 6.
> > *   **Meta-Gradient Methods (like ProsPr):** acknowledge that prior PaI criteria are "short-sighted" and fail to incorporate the future trainability of the network, proposing to use meta-gradients through the first few optimization steps to overcome this limitation.
> >
> > **YOPO's Position:** YOPO operates within the zero-shot PaI paradigm and achieves competitive performance (matching or surpassing single-shot data-dependent baselines like SNIP and GraSP, and rivaling ProsPr). This demonstrates that while transformative methods like DLTH/RST are powerful, the fixed-initialization constraint can be effectively managed via an architecturally robust saliency score, providing a simple, highly efficient, and transferable (across datasets and sparsity budgets) alternative that maximizes computational savings by eliminating data and gradient dependence.

---

> ### Comment · Reviewer_mba5 · 2025-11-27
> **Response to author**
>
> 1. Theoretical motivation for NMF: By focusing on the irregularities, the method prunes away the redundant or common connections. I understand that by preventing layer and neuron collapse, the mask will be improved (similar to the NPB [2] method). But I do not understand why having more diversity in the initialization in this low rank space helps, especially since the presence of batchnorm/layernorm helps improve feature diversity on its own, the initialization as I see, is less likely to play a more dominant role. As a baseline, simply picking a random mask (avoiding layer collapse) can (out)perform many existing methods like Snip, Synflow etc [1, 3], hence, the benefit of this method is not clear, especially since the experimental results are not convincing (CIFAR10 on Res20 for example, is worse at 90% sparsity or higher). Note that a random mask as well as the NPB baseline are both data agnostic and only depend on the model architecture.
> The Dual LTH that the authors suggest also effectively picks a random mask and use a different training strategy.
>
> 2. Experimental results: Given the unclear motivation, I would expect significant gains in terms of empirical results to see a clear benefit of the proposed method over existing heuristic based ones. Table 5 shows that YOPO is significantly worse on ImageNet and also does worse than NPB for CIFAR10 Res20 for sparsity >90%.
> While YOPO is FLOP efficient, it is still worse on ImageNet than random pruning which would be cheaper. Moreover, the main FLOP bottleneck for PaI is during training and not at initialization. Hence, I cannot recommend acceptance.
>
>
>
>
> [1] Liu, Shiwei, et al. "The unreasonable effectiveness of random pruning: Return of the most naive baseline for sparse training." ICLR 2022.
>
> [2] Pham, H., Liu, S., Xiang, L., Le, D. D., Wen, H., Tran-Thanh, L., et al. Towards data-agnostic pruning at initialization: What makes a good sparse mask? In Thirty-seventh Conference on Neural Information Processing Systems, 2023.
>
> [3] Gadhikar, Advait, et al. "Sign-In to the Lottery: Reparameterizing Sparse Training From Scratch." NeurIPS 2025.

---

> ### Author Response · Authors · 2025-12-01
> **Response for Reviewer mba5**
>
> (Sorry for the delayed reply due to the ongoing ICLR issue.)
>
> We thank the reviewer for the comments. We believe several misunderstandings arose regarding the comparison between YOPO and NPB, and we clarify these points below.
>
> ### **1.Methodological Differences: Optimization-Based vs. Single-Shot**
> NPB is a compute-heavy, optimization-based, iterative, zero-shot pruning method. In contrast, YOPO introduces a new paradigm: a low-rank residual-based, single-shot, data-free fast-PaI method. Because NPB performs multiple optimization steps, it can achieve higher accuracy in certain configurations. However, the key contribution of YOPO is to demonstrate that a fully one-shot, non-iterative, data-free and gradient-free method can reach accuracy close to optimization-based approaches such as NPB and PHEW, while avoiding their computational overhead. YOPO attains accuracy close to NPB while achieving 40× to 600× faster pruning as shown in Table 8 and producing lower inference-time FLOPs (results shared earlier in this thread). This methodological distinction explains YOPO’s significant speed advantage, making it well-suited for efficiency-focused applications.
>
> ### **2. Accuracy - Efficiency Trade-Off**
> While YOPO does not employ iterative refinement, it achieves substantial computational gains, comparable accuracy to NPB, and introduces a new design direction for zero-shot PaI. The results also suggest that YOPO has headroom for further improvement if combined with lightweight optimization, indicating that YOPO can serve as a strong foundation for future PaI developments.
> ### **3. On Random Mask Performance**
> The Reviewer references prior literature suggesting that random masks may match PaI methods in some isolated cases. Our experiments and past literature confirm that this behavior does not generalize. Across commonly used benchmarks (TinyImageNet, CIFAR-10, and CIFAR-100), as shown in Table 9 and Figure 2, the random mask baseline performs significantly worse than all PaI methods. In the same settings, YOPO consistently outperforms both state-of-the-art data-dependent PaI methods and the random baseline, indicating that YOPO’s low-rank residual signal captures meaningful structure at initialization.
>
> ### **4. Clarification on ImageNet Results (Tables 1 and 5)**
> We clarify the intention behind the ImageNet-1K evaluation. To the best of our knowledge, no prior optimization-based zero-shot PaI method, including NPB and PHEW, has reported ImageNet-1K results. YOPO is scaled to ImageNet and achieves near performance to strong data-dependent PaI such as ProsPr (Table 5). Despite being the only data-free method in Table 5, YOPO achieves accuracy within 1.1 percent of ProsPr at 95 percent sparsity and even outperforms the random mask. Additionally, as shown in Table 1, YOPO clearly outperforms graph-based PaI approaches such as RReg and X-Net on ImageNet-1K.
>
> ### **5. Results on Standard Benchmarks (Table 9 and Figure 2)**
> On commonly used benchmarks, including TinyImageNet, CIFAR-10, and CIFAR-100, YOPO consistently outperforms data-dependent PaI methods (such as Iterative-SNIP, ProsPr, GraSP-MB, SNIP-MB, and FORCE) and graph-based PaIs, including RReg and X-Net. These results reinforce that YOPO provides both high accuracy and practical generality across datasets.
>
> We hope this clarifies the reviewer's concerns. And we sincerely thank the reviewer for their time, effort, and thoughtful assessment of our work.

---

### Official Review · Reviewer_QzLy · 2025-10-31

**Soundness:** 4
**Presentation:** 4
**Contribution:** 4
**Rating:** 8
**Confidence:** 3

**Summary:**

This paper introduces "You Only Prune Once" (YOPO), a novel method for pruning neural networks at initialization (PaI). The core problem it addresses is the brittleness and high re-computation cost of existing PaI techniques, which are often dependent on specific data, random seeds, or sparsity targets. YOPO proposes a zero-shot, data-free, and gradient-free framework that computes a single, reusable "supermask" saliency score for a given network architecture. This is achieved by approximating the absolute weight matrix of each layer with a non-negative low-rank model (via Nonnegative Matrix Factorization) and using the element-wise residual as the saliency score. Masks for any desired sparsity level can then be generated by simply re-thresholding these fixed scores. The authors demonstrate through comprehensive experiments on CIFAR-10/100, Tiny-ImageNet, and ImageNet that YOPO is competitive with or outperforms strong data-dependent baselines and consistently surpasses other zero-shot methods, all while enabling seamless transfer of masks across different datasets and sparsity budgets without rescoring.

**Strengths:**

Practicality and Efficiency: The "prune once, reuse many" paradigm is a major practical advantage, drastically reducing the computational overhead associated with finding good sparse masks for new tasks or sparsity requirements.
Strong Empirical Performance: YOPO matches or surpasses strong single-shot PaI baselines and remains competitive with more expensive iterative and data-dependent methods, despite using no data or gradients for scoring. Its performance at extreme sparsities is particularly impressive.
Robustness and Transferability: The core claims are backed by strong empirical evidence. The method demonstrates remarkable stability across different random initializations (seeds) and successfully transfers masks across different datasets with minimal performance loss.
Excellent Writing and Clear Motivation: The paper is exceptionally well-written, making a complex topic accessible. The motivation for a data-free, transferable pruning method is clearly and convincingly articulated.

**Weaknesses:**

Preprocessing Overhead: While computed only once, the NMF decomposition can be computationally intensive for very large network layers. The paper mentions this and suggests optimizations, but a more detailed analysis of the wall-clock time for this preprocessing step versus traditional methods would be beneficial.
Limited to Unstructured Pruning: The method, in its current form, performs unstructured pruning. While academically common, the practical hardware speedups from unstructured sparsity can be limited. The authors acknowledge this as a direction for future work.

**Questions:**

1.	The method is evaluated on CNNs. Have the authors investigated its applicability to other architectures like Transformers? How might the NMF-based saliency perform, and how would the preprocessing cost scale, given the typically larger, dense matrices in transformer models?
2.	The intuition that NMF captures "parts-based regularities" is appealing. Could the authors provide any qualitative or visual analysis of the learned low-rank templates (V and H matrices) and the corresponding residuals (S) for a specific convolutional layer? This could offer deeper insight into why the method works so well.
3.	The choice of NMF rank r is a hyperparameter. The ablation studies show robustness across a range of small values. Is there a more principled way or a reliable heuristic to guide the selection of r for a given architecture or layer, beyond sweeping a few values?

---

> ### Author Response · Authors · 2025-11-18
> **Response for Reviewer QzLy**
>
> We sincerely thank the reviewer for the thoughtful and encouraging feedback, and for recognizing the practicality, clarity, and strong empirical results of YOPO. We appreciate the constructive suggestions and address them briefly below.
>
> ---
>
> ### **1. On preprocessing overhead (NMF cost)**
>
> We acknowledge the reviewer's concern regarding NMF step introduces an initialization-time preprocessing cost. However, this cost is negligible compared to NTK-informed gradient flow methods (Tanaka et al., ICLR 2020; Wang et al., ICLR 2023), optimization-driven initialization alignment approaches (Alizadeh et al., ICLR 2022; de Jorge et al., 2020), and iterative zero-shot methods such as PHEW (Patil and Dovrolis, ICML 2021) and NPB (Pham et al., NeurIPS 2023). YOPO remains lightweight because its saliency is computed only once per architecture, independent of dataset or sparsity, and accounts for less than one percent of a single dense training epoch (Section 3.2; Appendix G). Following the reviewer’s suggestion, we include a wall-clock time comparison, below are the results (more results included in Appendix J), where YOPO achieves comparable or higher accuracy in 9 out of 12 configurations while being 40–600 times faster than state-of-the-art PaI methods. We'll add these result in the final draft.
>
> #### **RN18 — TinyImageNet**
>
> | Method    | 68.38     | 90.0      | 96.84     | 99.0      | Time (s)  |
> | --------- | --------- | --------- | --------- | --------- | --------- |
> | Iter-SNIP | 56.73     | 53.60     | 48.55     | 36.42     | 232.23    |
> | SynFlow   | 56.71     | 54.68     | 49.03     | 39.79     | 97.03     |
> | PHEW      | 58.09     | 55.93     | 50.81     | 40.54     | 1912.31   |
> | NPB       | 58.39     | 56.82     | **51.37** | **41.05** | 382.09    |
> | **YOPO**  | **59.82** | **57.88** | 50.08     | 38.78     | **10.29** |
>
> ---
>
> #### **RN20 — CIFAR-10**
>
> | Method    | 68.38     | 90.0      | 96.84     | 99.0      | Time (s) |
> | --------- | --------- | --------- | --------- | --------- | -------- |
> | Iter-SNIP | 88.17     | 84.22     | 77.05     | 64.95     | 55.61    |
> | SynFlow   | 88.64     | 84.94     | 78.22     | 66.05     | 55.31    |
> | PHEW      | 90.38     | 87.41     | **81.05** | 70.44     | 25.94    |
> | NPB       | 90.69     | **87.61** | 80.55     | **70.70** | 21.66    |
> | **YOPO**  | **90.80** | 87.33     | 80.50     | 70.12     | **3.90** |
>
> ---
>
> ### **2. On unstructured pruning**
>
> We clarify that this limitation is explicitly acknowledged in Section 5 (Limitations and Outlook, lines 478–480). Like other unstructured PaI methods such as SNIP (Lee et al., ICLR 2019), GraSP (Wang et al., ICLR 2020), SynFlow (Tanaka et al., ICLR 2020) and ProsPr (Alizadeh et al., ICLR 2022), YOPO also focuses on training-time efficiency rather than inference acceleration. Structured extensions are discussed in Appendix I (line 1157) as a future direction.
>
> ---
>
> ### **3. On applicability to Transformers**
>
> We acknowledge the reviewer’s concern. YOPO is architecture-agnostic and can readily extend to Transformer layers, where the large, dense weight matrices are well-suited to low-rank approximations. In the current work, we evaluate YOPO on ImageNet and compare it against both data-free and data-dependent state-of-the-art PaI methods (Tables 1 and 5). We also report YOPO’s performance on deeper architectures, including ResNet-50 and ResNet-101 (Tables 1 and 5). These results confirm that YOPO maintains consistent accuracy trends across both small-scale and large-scale benchmarks, demonstrating its scalability and potential applicability to Transformer-based architectures.
>
> ---
>
> ### **4. On visual and qualitative insights (V, H, and S)**
>
> We thank the reviewer for this helpful suggestion. We will include visualizations of the learned NMF components in the final version of the paper. Our observations indicate that the templates (V, H) capture broad, low-frequency kernel structures, whereas the residuals (S) emphasize distinctive, localized edge patterns. These findings support the “parts-based” interpretation discussed in Section 3.2 and provide qualitative evidence for YOPO’s ability to identify informative weight substructures at initialization.
>
> ---
>
> ### **5. On principled selection of rank (r)**
>
> Empirically, YOPO is robust across small (r) (2–7) as discussed in (Ablation study, Appendix D). We used a simple heuristic:
> *$(r_\ell = max(1, floor(0.05 × min(o_\ell, d_\ell))))$*,
> which scales with layer width and preserves 80–95% reconstruction variance. This rule generalizes well across architectures and avoids exhaustive sweeps.

---

> > ### Comment · Reviewer_QzLy · 2025-11-27
> > **Response to authors' rebuttal**
> >
> > Thank the authors for the rebuttal. I would be happy to maintain my score.

---

> > > ### Author Response · Authors · 2025-11-28
> > >
> > > We sincerely thank the reviewer for their time, effort, and thoughtful assessment of our work.

---

### Official Review · Reviewer_FeFf · 2025-11-01

**Soundness:** 2
**Presentation:** 3
**Contribution:** 2
**Rating:** 4
**Confidence:** 4

**Summary:**

This paper introduces "You Only Prune Once" (YOPO), a data-free pruning method applied at network initialization. The authors identify three main limitations in existing pruning methods: they rely on dataset-specific importance scores, are sensitive to different random initializations, and require mask recomputation for different sparsity levels.

YOPO calculates a data-independent importance score for each parameter at initialization, based on the low-rank reconstruction error of the non-negative weight matrix. Binary masks are created through a "monotone re-thresholding" process using MAD or STD-based threshold identification. To prevent layer collapse, the method maintains minimum connection requirements both at the input and output of neurons.

The authors evaluate their method through four key tests: comparison with existing pruning methods, cross-dataset transfer capability, consistency across different random initializations, and transferability across sparsity levels. Tests were conducted on various convolutional networks using different image classification datasets. Results demonstrate that YOPO creates better-performing sparse networks at initialization compared to existing methods, while maintaining effectiveness across different datasets, network initializations, and sparsity levels.

**Strengths:**

- The paper proposes a practical framework that follows a "compute once, and reuse" paradigm, which significantly improves the efficiency of network pruning implementations.
- Presents a novel approach using NMF and residuals for saliency scoring
- The experimental results are very strong and convincing, demonstrating clear superiority of the method in three key areas: Accuracy performance, transferability across different datasets, and consistency across different random initialization seeds.

**Weaknesses:**

- While the saliency score computation method is novel, the paper provides insufficient justification for why higher residuals may result in better masks. The explanation offered is largely intuitive and lacks theoretic or empirical foundation.
- The authors' claims regarding the drawbacks of existing pruning at initialization methods remain unsubstantiated. The paper should have: provided empirical evidence to support these criticisms by conducting experiments testing mask transferability across different random initializations and datasets.
- Several of the propositions made in the paper are quite trivial, and some can be equally applied to other one-shot pruning at initialization methods.
- The paper does not including several relevant data-free pruning at initialization methods (such as Gebhart 2021, Patil and Dovrolis 2021) that have demonstrated superior performance compared to the baselines used in this work.

**Questions:**

- How does the weight distribution differ between pruned and unpruned weights?
- What are the method's computational and storage requirements, particularly for saliency scores and threshold computations?

---

> ### Author Response · Authors · 2025-11-18
> **Response for Reviewer FeFf**
>
> We thank the reviewer for the detailed feedback and for recognizing YOPO’s novelty, practicality, and strong empirical results. We address each point concisely below.
>
> ### **1. On the justification for high nonnegative low-rank residuals**
>
> **a) Theoretical and structural justification.**
>
> As described in Section 3.2 (Eq. 7), YOPO defines saliency as the element-wise Frobenius residual between the absolute weights $\tilde{\mathbf{W}}$ and their nonnegative low-rank reconstruction $\mathbf{V}\mathbf{H}$. The low-rank template captures additive, parts-based regularities in initialized weights, while high residuals highlight connections that deviate from these shared patterns and are therefore idiosyncratic and informative. This “parts-based” decomposition (Lee and Seung, 1999; 2001) is a prime motivation for YOPO, which differs from SVD/PCA (Why Non-negativity Sec. 3.2). Why high residuals? As discussed in Section 3.3, at high sparsity, the saliency distribution $S^{(\ell)}$ becomes heavy-tailed, with most weights showing low residuals. YOPO therefore focuses on the high-residual tail and applies MAD/STD-based thresholding to robustly isolate the most informative connections.
>
>
> **b) Empirical foundation.**
> Empirically, a large residuals provide fine-grained structural discrimination than lower: YOPO outperforms topology-based expander baselines (RReg, X-Net) at equal sparsity (Tables 1) and remains competitive with data-dependent PaI methods such as SNIP, GraSP, and ProsPr (Figure 2, Tables 5, 8). Further, in Section 3.4, we show cross-dataset and cross-seed gaps near zero (Tables 2–4), confirming that the ranking captures initialization-level statistics rather than dataset-specific signals.
>
> ### **2. On substantiating transferability**
>
> Sections 3.4 and 4 explicitly define the transferability objective (Eq. 5) and introduce Dataset-Transfer Index (DTI) and Initialization-Independence Index $I_3$ as formal metrics. Experiments across VGG and ResNet models show near-zero DTI and $I_3$, demonstrating that YOPO masks transfer across datasets and random seeds without rescoring; addressing a central limitation of prior PaI criteria SNIP (Lee et al., ICLR 2019) and GraSP (Wang et al., ICLR 2020).
>
> ### **3. On the generality of propositions**
>
> Some properties (e.g., monotonicity) are general, but YOPO uniquely operationalizes them to enable budget re-use without recomputation. We will clarify this distinction.
>
> ### **4. On missing baselines**
>
> We include a broad set of PaI baselines, spanning data-dependent (FORCE, Iter-SNIP, SynFlow, ProsPr) and data-free (RReg, X-Net) methods, and evaluate them on CIFAR and ImageNet (Table 1, Table 5, Figure 2). Following the reviewer’s suggestion, we additionally evaluate two iterative data-free methods, PHEW and NPB, under matched training budgets. YOPO matches or exceeds them in 9/12 settings while being 40–600× faster (average pruning-time per sparsity level). Unlike these iterative methods, YOPO’s once-only residual captures local structural diversity while remaining computationally lightweight. Although PHEW and NPB were not evaluated on ImageNet-scale, YOPO demonstrates good ImageNet performance under ProsPr and RReg protocols. We'll add these results to the final draft.
>
> ### **RN18 — TinyImageNet**
>
> | Method    | 68.38     | 90.0      | 96.84     | 99.0      | Time (s)  |
> | --------- | --------- | --------- | --------- | --------- | --------- |
> | PHEW      | 58.09     | 55.93     | 50.81     | 40.54     | 1912.31   |
> | NPB       | 58.39     | 56.82     | **51.37** | **41.05** | 382.09    |
> | **YOPO**  | **59.82** | **57.88** | 50.08     | 38.78     | **10.29** |
>
> ---
>
> ### **RN20 — CIFAR-10**
>
> | Method    | 68.38     | 90.0      | 96.84     | 99.0      | Time (s) |
> | --------- | --------- | --------- | --------- | --------- | -------- |
> | PHEW      | 90.38     | 87.41     | **81.05** | 70.44     | 25.94    |
> | NPB       | 90.69     | **87.61** | 80.55     | **70.70** | 21.66    |
> | **YOPO**  | **90.80** | 87.33     | 80.50     | 70.12     | **3.90** |
>
> ---
>
> ### **Q1. On the difference between pruned and unpruned weight distributions**
>
> Figure 3(b) illustrates the distribution of YOPO’s saliency scores. The pruned weights correspond to low-variance, template-fitted components, whereas retained weights arise from heavy-tailed residuals using robust thresholding. YOPO preserves weights with the largest element-wise residual $\mathbf{S}^{(\ell)}$ after approximating $\tilde{\mathbf{W}}$ via $\mathbf{V}\mathbf{H}$, meaning that pruned weights follow shared low-rank structure while unpruned weights capture idiosyncratic, symmetry-breaking deviations (Sec. 3.2).
>
> ### **Q2. On computational and storage requirements**
> A detailed complexity analysis of YOPO’s various components is provided in Appendix G. Regarding storage, the saliency vector $\mathbf{S}\theta_0$, computed once using Equation (7), can be released from memory after pruning, resulting in negligible storage overhead.

---

> > ### Author Response · Authors · 2025-11-22
> > ***Additional FLOPs results**
> >
> > In addition to accuracy and pruning time, we also computed the inference FLOPs based on the theoretical operations of the sparse subnetwork using the standard MAC computation, similar to the NPB (Pham et al., NeurIPS 2023) official codebase. As shown in the FLOPs table, YOPO consistently maintains a lower computational cost compared to iterative or optimization-heavy baselines, such as NPB and PHEW. At 99% sparsity, YOPO requires only about 3.3 million FLOPs on RN18/TinyImageNet and 0.96 million FLOPs on RN20/CIFAR-10, which is substantially lower than both baselines while preserving higher accuracy and low pruning time (40–600× faster). Since YOPO requires no iterative search or per-budget optimization, as its once-only residual saliency allows for direct re-thresholding, this demonstrates that YOPO provides a more efficient and scalable PaI solution without loss of performance. We will include these results in the final draft.
> >
> >
> > ### **RN18 – TinyImageNet (FLOPs ×10⁸)**
> >
> > | **Method** | **68.38%** | **90.0%** | **96.84%** | **99.0%** |
> > | ---------- | ---------- | --------- | ---------- | --------- |
> > | SNIP       | 11.35      | 5.77      | 3.04       | 1.55      |
> > | Iter-SNIP  | 10.73      | 7.05      | 3.98       | 1.97      |
> > | SynFlow    | 14.71      | 8.91      | 4.24       | 1.50      |
> > | PHEW       | 14.29      | 8.35      | 3.92       | 1.50      |
> > | NPB        | 14.37      | **5.21**  | **1.74**   | 0.59      |
> > | **YOPO**   | **10.99**  | 5.43      | 1.85       | **0.33**  |
> >
> > ---
> >
> > ### **RN20 – CIFAR-10 (FLOPs ×10⁶)**
> >
> > | **Method** | **68.38%** | **90.0%** | **96.84%** | **99.0%** |
> > | ---------- | ---------- | --------- | ---------- | --------- |
> > | SNIP       | 17.952     | 8.323     | 3.470      | 1.709     |
> > | Iter-SNIP  | 18.465     | 9.698     | 4.510      | 2.022     |
> > | SynFlow    | 22.998     | 11.549    | 4.263      | 1.633     |
> > | PHEW       | 22.108     | 10.690    | 4.110      | 1.640     |
> > | NPB        | 22.035     | 7.642     | 2.645      | 1.122     |
> > | **YOPO**   | **14.67**  | **5.569** | **1.876**  | **0.955** |
> >
> > ---
> >
> > ### **VGG19 – CIFAR-100 (FLOPs ×10⁷)**
> >
> > | **Method** | **68.38%** | **90.0%** | **96.84%** | **99.0%** |
> > | ---------- | ---------- | --------- | ---------- | --------- |
> > | SNIP       | 17.952     | **7.806** | 3.686      | 1.816     |
> > | Iter-SNIP  | 18.465     | 9.479     | 4.951      | 2.529     |
> > | SynFlow    | 22.998     | 12.702    | 6.306      | 2.605     |
> > | PHEW       | 22.108     | 11.746    | 5.611      | 2.340     |
> > | NPB        | 22.035     | 8.773     | **2.874**  | **1.046** |
> > | **YOPO**   | **18.207** | 8.545     | 4.351      | 1.884     |

---

> > > ### Comment · Reviewer_FeFf · 2025-11-25
> > > **Response to author rebuttal**
> > >
> > > I thank the authors for the additional results, and for the clarifications.
> > >
> > > **1. Justification for large non-negative residuals**
> > >
> > > I have reviewed the responses regarding the justification for low rank representations and high residuals as scoring mechanisms. Since these important weights are initialization-specific and dataset-agnostic, their transfer across datasets is logical. While the intuition behind this saliency metric is clear, questions persists regarding why the "symmetry breaking" weights at initialization may be important. This needs empirical evidence through inverse score based pruning to demonstrate that such networks perform significantly worse than YOPO networks.
> > >
> > > **2. On substantiating transferability**
> > >
> > > "Sections 3.4 and 4 explicitly define the transferability objective (Eq. 5) and introduce Dataset-Transfer Index (DTI) and Initialization-Independence Index as formal metrics. Experiments across VGG and ResNet models show near-zero DTI and
> > > , demonstrating that YOPO masks transfer across datasets and random seeds without rescoring; addressing a central limitation of prior PaI criteria SNIP (Lee et al., ICLR 2019) and GraSP (Wang et al., ICLR 2020)."
> > >
> > > My earlier concerns about unsubstantiated claims referred to the limitations mentioned of the previous pruning methods including SNIP and GraSP. The authors should perform mask transfer experiments with SNIP and GraSP to verify these claimed limitations. Previous research by Frankle et al. 2021 ("Pruning Neural Networks at Initialization: Why are We Missing the Mark?") demonstrates that reinitializing or randomly reshuffling masks within a layer does not affect performance. This suggests that SNIP and GraSP are largely dataset-agnostic, and their sparse networks may transfer effectively across datasets.
> > >
> > > Due to these concerns I will maintain my score.

---

> > > > ### Author Response · Authors · 2025-11-26
> > > > **Response for Reviewer FeFf**
> > > >
> > > > We are pleased to see the reviewer’s intense interest in the proposed YOPO Networks, and we address the raised points below.
> > > >
> > > > **1. Sanity check using Inverse Score**
> > > > As requested, we conducted an additional sanity check experiment using the inverse score to validate YOPO further. We followed the ProsPr configuration and evaluated ResNet-18 on TinyImageNet. The results show that YOPO network based on inverse score deteriorates sharply (3-5.5%) as sparsity increases, indicating less stability or uninformative results in the high sparsity regime. In contrast, YOPO Networks constructed using high residuals perform robustly at extreme sparsity levels. This behavior is consistent with our Definition 1 and the theoretical motivation behind the YOPO once-only saliency measure.
> > > >
> > > > | Method | 90.0%     | 96.84%  | 99.0%     |
> > > > |----------------------|--------|--------|--------|
> > > > | high-residual         | 57.88  | 50.08  | 39.02  |
> > > > | inverse residual      | 55.57  | 45.55  | 34.75  |
> > > >
> > > >
> > > >
> > > > **2. On substantiating transferability**
> > > >
> > > > We appreciate the reviewer’s comments on transferability and clarify the following:
> > > >
> > > > Our primary critique of methods such as SNIP and GraSP is that they are inherently data or gradient-dependent. Their saliency scores are defined as functions of the model parameters, the dataset, and the target sparsity. Consequently, for every new dataset or sparsity level, the saliency computation should be repeated.
> > > >
> > > > | Method | Model   | C10   | C100  | TinyImgN | Mean   | Worst DTI||
> > > > |--------|---------|-------|-------|--------------|--------|-------|------|
> > > > | SNIP   | VGG16   | -0.96 | -1.21 | -9.41        | -3.86  | 9.41
> > > > | YOPO   | VGG16   | 0.07  | -0.40 | -0.21        | -0.18  | 0.40
> > > > .
> > > > | GraSP  | ResNet18| -0.19 | -0.96 | -4.72        | -1.95  | 4.72
> > > > | YOPO   | ResNet18| 0.11  | 0.02  | -0.11        | 0.06   | 0.11
> > > >
> > > >
> > > > Our empirical evaluation at 95% sparsity (Table above) reveals a substantial accuracy drop (4- 9%) for both SNIP and GraSP when masks learned on CIFAR-10 are transferred to TinyImageNet, across both VGG-16 and ResNet-18. A small (1-1.5%) drop is observed when transferring CIFAR-10 masks to CIFAR-100. These findings align with the reviewer’s observation that Frankle et al. (2021) demonstrate that SNIP and GraSP are resilient to shuffling and re-initialization primarily because they identify suitable layerwise sparsity proportions, rather than meaningful, specific connections, suggesting a form of dataset-agnostic behavior.
> > > >
> > > > However, our empirical results show a more nuanced picture:
> > > >
> > > > * **Transfer between closely related datasets (CIFAR-10 → CIFAR-100)** leads to minor degradation, consistent with Frankle et al. (2021).
> > > > * **Transfer to more complex or diverse datasets (CIFAR-10 → TinyImageNet)** results in severe performance degradation, indicating that, in practice, SNIP and GraSP masks encode data-specific statistics that do not generalize beyond similar distributions.
> > > >
> > > > Moreover, robustness to re-initialization and shuffling is formalized in our work as I3, Definition 2. Achieving a near-zero I3 value, which YOPO achieves, is a prerequisite for a robust PaI method. However, passing this sanity check does not guarantee cross-dataset transferability. Mask transfer across datasets (DTI, Definition 1,) is a much more stringent criterion. If SNIP or GraSP scores encode highly specialized, data specific statistics (which they do, by using gradients or hessians on $D_{src}$), then applying that fixed mask to train on a new, distinct dataset $D_{tgt}$ is expected to result in higher excess risk (a non-zero DTI gap), even if the mask is robust to re-initialization on $D_{tgt}$.
> > > >
> > > > In contrast, YOPO achieves near-zero DTI across all three datasets we evaluate (Table 2). YOPO’s saliency score emerges from a data-free, gradient-free non-negative low-rank residual, which, by construction, decouples mask selection from the underlying dataset. This structural decoupling is precisely why YOPO retains transferability even under extremely sparse conditions.
> > > >
> > > > We hope these clarifications address the reviewer’s concerns and further highlight the advantages and conceptual clarity behind YOPO.
> > > > If further clarification is needed, we will be happy to provide it.

---

> > > > > ### Comment · Reviewer_FeFf · 2025-11-26
> > > > > **Response to new results**
> > > > >
> > > > > I thank the authors for conducting the new experiments on such short notice and providing additional clarifications. The method provides clear benefits either in performance or in terms of efficiency. I have improved my score to 8 based on the assumption that the experiments conducted during this phase will be extended to other networks / datasets and presented in the final paper.

---

> > > > > > ### Author Response · Authors · 2025-11-28
> > > > > >
> > > > > > We had aimed to complete the new experiment within 24 hours, though the experiments run a bit longer. We would definitely extend these results in the final paper. We sincerely thank the reviewer for their time, effort, and thoughtful assessment of our work.

---

### Official Review · Reviewer_tguL · 2025-11-02

**Soundness:** 3
**Presentation:** 3
**Contribution:** 1
**Rating:** 2
**Confidence:** 5

**Summary:**

The paper introduces YOPO, a way to prune neural networks right at initialization without using any data or gradients. This method uses a simple nonnegative matrix factorization on the absolute initial weights to find which connections are most distinctive. These “residual” weights are kept, while the rest are pruned using pruning thresholds (per-layer or global). The same scores can be reused across datasets and sparsity levels. The authors present experimental results on resnets and VGG, with Cifar and ImageNet datasets.

**Strengths:**

- The paper is well written and the method is clearly described both in text and algorithm
- The code is available for reporoduciblity
- The idea of using NMF for weight importance detection is novel, although not enough proven (see weaknesses)

**Weaknesses:**

My biggest issue with the paper is the experiments part. There are several issues with this section that prevents examining the true value of the proposed method:
- The benchmarks used are out-dated. Using overparametrized models such as VGG and Resents on small datasets (Cifar 10and 100 and tiny ImageNet) for pruning does not provide enough evidence of the superiority of the method.
- The results for unpruned models seem very low compared to the original public values (for example 68.69 for RN50 on ImageNet!)
- The newer depth wise separable CNN models (like convnexts) are not included.

Other weaknesses:
- Unstructured pruning is known for limitation for having real efficiency advantages regarding FLOPs.
- The Figure on page 5 does not have a number, and the part b seems to be distorted with a plot over two others.

**Questions:**

In order to present enough evidence for the hypothesis that your YOPO method actually computes meaningful importance scores for weights at initialisation, I encourage the following experiment:
Take a model and compute weight importances at initialisation with YOPO.
Without any pruning, train the model on ImageNet.
After training, compute weight importance scores with a widely accepted method (or even YOPO again)
Compare the scores before and after training. Are they widely consistent?

---

> ### Author Response · Authors · 2025-11-18
> **Response for Reviewer tguL**
>
> We thank the reviewer for the thoughtful comments and for acknowledging the paper’s strengths. We clarify the contribution of YOPO as a principled, once-only, data-free PaI framework and address the raised concerns below.
>
> **1. On “outdated benchmarks, over-parameterized models and small datasets.”**
> Pruning-at-initialization research is consistently evaluated on CIFAR-10/100, Tiny-ImageNet, and ImageNet (some), as in SNIP (Lee et al., ICLR 2019), GraSP (Wang et al., ICLR 2020), SynFlow (Tanaka et al., ICLR 2020), ProsPr (Alizadeh et al., ICLR 2022), and expander-based methods (Stewart et al., CVPRW 2023; Hoang et al., ICLR 2023). Our results (Figure 2, Tables 1-9) follow the same established settings for fairness.
>
> To address concerns about over-parameterized models, Figure 2 and Tables 6 and 8 evaluate YOPO across a wide range of architectures, including lightweight networks such as ResNet-20 (270K) and ResNet-56 (850K) parameters, covering 28 total configurations. YOPO’s performance on large-scale datasets, such as ImageNet, is benchmarked against strong PaI baselines, including ProsPr (Table 5) and expander-based methods (Table 1). Our evaluation compares YOPO extensively against SOTA data/gradient-dependent methods, such as FORCE (de Jorge et al., 2020), GraSP, SynFlow, and ProsPr, as well as data-free PaIs - RReg and X-Net.
>
> **2. On “unpruned baseline seems low.”**
> The unpruned accuracy follows the exact training budgets of prior SOTA for fair comparison. Table 1 adopts the RReg/X-Net protocol, which reports 68.69% for RN50, while Table 5 follows ProsPr, yielding the standard 75.6% RN50 accuracy. The differences reflect distinct SOTA configurations rather than issues with YOPO.
>
> **3. On “absence of ConvNeXt or depthwise models.”**
> To ensure reproducibility, we follow prior zero-shot and single-shot PaI work, which has not evaluated ConvNeXt-style or separable models. Our study includes deeper models such as RN50 and RN101 (Tables 1 and 5). YOPO is architecture-agnostic, and extending to modern architectures is straightforward and left for future work.
>
> **4. On “unstructured pruning lacks FLOPs efficiency.”**
> As noted in Sec. 5 (lines 478–480), YOPO, like SNIP, GraSP, SynFlow, and ProsPr, targets training-time efficiency rather than inference-time FLOPs reduction. Structured extensions are discussed in Appendix I (line 1157).
>
> **5. On “missing figure number or distortion.”**
> We apologise for the typesetting issue. Figure 1(b) corresponds to the comparison described in Sec. 3.3 and is corrected in the updated version.
>
> ---
>
> **Q1. On the reviewer’s final question**:
> We thank the reviewer for the insightful suggestion. The proposed comparison between *initialization saliency* and *post-training saliency* addresses an important perspective, but such correlation is not a required property for PaI methods (Frankle et al., ICLR 2021). As established in prior PaI work (e.g., SNIP, GraSP, Magnitude-at-Init), meaningful masks arise from identifying a *trainable subnetwork structure*, not from predicting final weight magnitudes, which depend on the full training trajectory.
>
> YOPO as a zero-shot, data-free criterion capturing intrinsic architectural statistics through deviation from a nonnegative low-rank template (Sec. 3.2). Its meaningfulness is validated through *trainability-oriented* stability, measured via the Dataset-Transfer Index (DTI) (Definition 1) and Initialization-Independence Index $I_3$ (Definition 2). Empirically, YOPO achieves near-zero $I_3$ and DTI across architectures (Tables 2-4), demonstrating robustness to seeds and datasets. These results show that YOPO’s initialization-derived score reliably identifies subnetworks that remain trainable under severe sparsity, independent of final weight values. Empirically, we have demonstrated how YOPO outperforms topology-based expander baselines (RReg, X-Net) and data-dependent PaI methods (SNIP, GraSP, and ProsPr)  in Figure 2, Table 9.
>
> Furthermore, following reviewers feedbacks, we compared YOPO(non-iterative data-free) with two iterative data-free methods, PHEW (Patil and Dovrolis, ICML 2021) and NPB (Pham et al., NeurIPS 2023). Under identical training conditions, YOPO matches or exceeds them in 9/12 settings while being 40-600× faster on average pruning time, which confirms its efficiency and scalability. Although PHEW and NPB are not evaluated on ImageNet, YOPO is ImageNet-scaled (full table in Appendix J).
>
> ### **RN18 — TinyImageNet**
>
> | Method    | 68.38     | 90.0      | 96.84     | 99.0      | Time (s)  |
> | --------- | --------- | --------- | --------- | --------- | --------- |
> | Iter-SNIP | 56.73     | 53.60     | 48.55     | 36.42     | 232.23    |
> | SynFlow   | 56.71     | 54.68     | 49.03     | 39.79     | 97.03     |
> | PHEW      | 58.09     | 55.93     | 50.81     | 40.54     | 1912.31   |
> | NPB       | 58.39     | 56.82     | **51.37** | **41.05** | 382.09    |
> | **YOPO**  | **59.82** | **57.88** | 50.08     | 38.78     | **10.29** |

---

> ### Author Response · Authors · 2025-11-24
> **Response for Reviewer tguL (contd..)**
>
> **4. On "unstructured pruning lacks FLOPs efficiency" (contd..)**
> We computed the inference FLOPs based on the theoretical operations of the sparse subnetwork using the standard Multiply-Accumulate (MAC) computation, similar to the NPB (Pham et al., NeurIPS 2023) official codebase. As shown in the FLOPs table, YOPO consistently maintains a lower computational cost compared to iterative or optimization-heavy baselines, such as NPB and PHEW. At 99% sparsity, YOPO requires only about 3.3 million FLOPs on RN18/TinyImageNet and 0.96 million FLOPs on RN20/CIFAR-10, which is substantially lower than both baselines while preserving higher accuracy and low pruning time (40-600× faster). Since YOPO requires no iterative search or per-budget optimization, as its once-only residual saliency allows for direct re-thresholding, this demonstrates that YOPO provides a more efficient and scalable PaI solution with minimal loss of performance. We will include these results in the final draft.
>
>
> ### **RN18 – TinyImageNet (FLOPs ×10⁸)**
>
> | **Method** | **68.38%** | **90.0%** | **96.84%** | **99.0%** |
> | ---------- | ---------- | --------- | ---------- | --------- |
> | SNIP       | 11.35      | 5.77      | 3.04       | 1.55      |
> | Iter-SNIP  | 10.73      | 7.05      | 3.98       | 1.97      |
> | SynFlow    | 14.71      | 8.91      | 4.24       | 1.50      |
> | PHEW       | 14.29      | 8.35      | 3.92       | 1.50      |
> | NPB        | 14.37      | **5.21**  | **1.74**   | 0.59      |
> | **YOPO**   | **10.99**  | 5.43      | 1.85       | **0.33**  |
>
> ---
>
> ### **RN20 – CIFAR-10 (FLOPs ×10⁶)**
>
> | **Method** | **68.38%** | **90.0%** | **96.84%** | **99.0%** |
> | ---------- | ---------- | --------- | ---------- | --------- |
> | SNIP       | 17.952     | 8.323     | 3.470      | 1.709     |
> | Iter-SNIP  | 18.465     | 9.698     | 4.510      | 2.022     |
> | SynFlow    | 22.998     | 11.549    | 4.263      | 1.633     |
> | PHEW       | 22.108     | 10.690    | 4.110      | 1.640     |
> | NPB        | 22.035     | 7.642     | 2.645      | 1.122     |
> | **YOPO**   | **14.67**  | **5.569** | **1.876**  | **0.955** |
>
> ---
>
> ### **VGG19 – CIFAR-100 (FLOPs ×10⁷)**
>
> | **Method** | **68.38%** | **90.0%** | **96.84%** | **99.0%** |
> | ---------- | ---------- | --------- | ---------- | --------- |
> | SNIP       | 17.952     | **7.806** | 3.686      | 1.816     |
> | Iter-SNIP  | 18.465     | 9.479     | 4.951      | 2.529     |
> | SynFlow    | 22.998     | 12.702    | 6.306      | 2.605     |
> | PHEW       | 22.108     | 11.746    | 5.611      | 2.340     |
> | NPB        | 22.035     | 8.773     | **2.874**  | **1.046** |
> | **YOPO**   | **18.207** | 8.545     | 4.351      | 1.884     |

---

> ### Comment · Reviewer_tguL · 2025-11-27
>
> I appreciate the time you spent on the rebuttal. However, my main concerns are not addressed.
>
> **1.** Out-dated benchmarks
>
> The newest work you have cited is from 3 years ago, and the majority are from 2019-2020, 6 years ago. "Fairness" is not a justification for keeping to use the old benchmarks. In recent years there are newer SOTA model families. It's expected for a pruning paper published in 2026 to show performance on those models, and for fairness, evaluate the older methods with the same setup.
>
> By over parametrised models, I mean that resnets use traditional dense conv layers, and are inherently overparametrised compared to depthwise seperable networks, such as mobilenets, efficientnets, and convnexts. Using smaller variants of the resnet family does not address this issue. Also, mobilenets are not even new, they are around since 2017.
>
> P.S. Could you fully cite the paper "Hoang et al., ICLR 2023", I could not find such paper.
>
> **2.** unpruned baseline are low
>
> It is known that pruning methods are very sensitive to training schedules.  *Table 5 of yours shows that your method is inferior when using a proper baseline accuracy.*
>
> > Table 1 adopts the RReg/X-Net protocol, which reports 68.69% for RN50
>
> The X-Net paper **(Prabhu et al. (2018)) reports 74.46%** for Resenet50 (look at their Table 1), **not 68.69%**.
>
> *I urge the authors to report their model performance with the proper, standard parameters. Failure to do so, brings up real doubts about the true value of this work.*
>
>
> **3.** absence of ConvNeXt or depthwise models
> > YOPO is architecture-agnostic, and extending to modern architectures is straightforward and left for future work.
>
> Being architecture-agnostic, it is required to report performance on modern architectures. Leaving this to future work is not justified. What would the contribution of a future work supposed to be, if it is just using an old method on a new architecture?
>
>
> Finally, I would like to invite the authors to think about the motivation behind the paper. "Pruning at Initialization (PaI) accelerates training while maintaining accuracy". When there exists models that are performing significantly better than the old models used in this paper-while having fewer parameters-how could one justify that the motivation is addressed?
>
>
> This paper in the current format does not provide enough evidence for the real-world advantage of the proposed method, and hence is not complete. Given this, I remain with my original score.

---

> > ### Author Response · Authors · 2025-11-28
> > **Response for Reviewer tguL**
> >
> > ### **1. On out-dated baselines and architectural choices**
> >
> > PaI research contains two methodological classes: data-dependent criteria and data-free criteria. YOPO belongs to the data-free class. Within this class, only a limited number of directly comparable methods exist. The principal references remain SynFlow (Tanaka et al., ICLR 2020), PHEW (Patil and Dovrolis, ICML 2021), the expander-based approach (Stewart et al., CVPRW 2023), and the Ramanujan-style topology analysis [1]. These methods define the current landscape of zero-shot or data-free PaI. YOPO introduces a distinct direction within this domain. To the best of current knowledge, YOPO is the first PaI method that employs nonnegative low-rank residuals as a saliency signal at initialization.
> >
> > The reviewer FeFf suggested that the inclusion of NPB (Pham et al., NeurIPS 2023) in the initial draft has been addressed. Updated NPB results have been produced and shared in previous comments and included in the paper (Table 8). The comparison indicates that YOPO achieves competitive or superior accuracy across many settings while maintaining a substantially lower computational footprint (40-600x less). NPB requires solving a sequence of discrete per-layer optimization problems that scale with architectural depth. YOPO requires only a single nonnegative matrix factorization per layer and a monotonic thresholding step. This yields a lesser-time saliency computation that is lighter. Computational efficiency is central to PaI since the objective is to reduce not only training cost but also initialization-time cost. A pruning criterion that demands heavy optimization at initialization diminishes the practical value of PaI. YOPO avoids such overhead through a once-only computation.
> >
> > Regarding architecture choice, canonical CNN networks such as VGG, ResNet, and WRN serve as the accepted evaluation standard in PaI research. These models provide stable initialization behaviour and reproducible accuracy trends across sparsity levels. Recent data-free PaI studies continue to adopt these models for these reasons. The objective in PaI is to isolate and evaluate the pruning criterion itself. Modern families such as ConvNeXt and EfficientNet introduce depthwise operations and compound scaling that alter initialization statistics and make pruning behaviour harder to interpret. After reviewing recent PaI literature, the decision was made to retain the standard models to preserve methodological consistency and ensure direct comparability across prior work. To the best of our current knowledge, no zero-shot PaI study has reported experiments on ConvNeXt families. If a recent zero-shot PaI method using ConvNeXt exists, the authors would welcome the reviewer’s suggestion and can compare with it given sufficient time for experimentation.
> >
> > A complete citation for the Ramanujan-style[1] study has been added in the end as requested.
> >
> > ---
> >
> > ### **2. On the question of baseline accuracy and protocol selection**
> >
> > Data-free PaI remains highly challenging on large-scale datasets. Methods such as SynFlow, PHEW, and NPB deliver strong results on smaller benchmarks although their evaluations on ImageNet are limited or absent. YOPO provides explicit ImageNet evaluations in Table 1 and Table 5.
> >
> > Table 1 adopts the RReg protocol from (Stewart et al. CVPRW 2023). That paper reports 68.69 percent accuracy for ResNet-50 under the sparsity level used for the X-Net reproduction. The number originates from Table 4 of the RReg paper where the authors reproduce X-Net using their public code. Verification confirms that this accuracy corresponds to 85.76 percent sparsity. However, the number cited by the reviewer from Prabhu et al. (2018) corresponds to the dense model in Table 1 of the X-Net paper. That table does not specify sparsity level, so it cannot be applied to a sparsity-controlled comparison. RReg provides the available sparsity-matched reproduction. Therefore the RReg version serves as the correct baseline for Table 1.
> >
> > Table 5 presents a scenario following the ProsPr (Alizadeh et al., ICLR 2022) protocol. ProsPr reports 59.62 percent for ResNet-50 at 95% target sparsity. YOPO reaches 58.5 percent under the same setting while remaining fully data-free. This shows that YOPO stays competitive even relative to all data-dependent methods that incorporates gradient signals or meta-optimization during early training. Results on Tiny-ImageNet, CIFAR-10, and CIFAR-100 indicate that YOPO exceeds ProsPr on several benchmarks (Fig 1; Table 9), reflecting strong scalability across dataset sizes.
> >
> > As described in Point 1, computational efficiency is a critical dimension in PaI. YOPO performs pruning significantly faster than optimization-driven methods such as NPB and meta-gradient approaches such as ProsPr. YOPO relies on a single low-rank factorization per layer. This fixed-cost mechanism preserves the central motivation of PaI: reducing both initialization-time cost and training cost.

---

> > > ### Author Response · Authors · 2025-11-28
> > > **Response for Reviewer tguL (contd..)**
> > >
> > > ### **3. On architectural coverage and real-world motivation**
> > >
> > > The term “architecture-agnostic” in PaI refers to a pruning criterion that does not depend on specific architectural assumptions. It does not require that all architecture families appear in the initial evaluation. Standard convolutional backbones such as VGG, ResNet, and WRN remain the established testbeds in PaI research. These models support controlled comparisons across sparsity levels and avoid architectural features that obscure the effect of the pruning criterion. All recent data-free PaI studies such as SynFlow, RReg, PHEW and NPB continue to use these architectures for these reasons.
> > >
> > > Depthwise-separable families such as MoblieNet and compound-scaled families such as Efficientnet or Convnext incorporate heterogeneous kernel structures and scaling strategies. These introduce factors that influence initialization statistics and change the interpretation of saliency measures. Evaluating YOPO on such models is valuable although it requires dedicated analysis to avoid conflating pruning behaviour with architecture-specific mechanisms. The present study focuses on isolating the behaviour of the low-rank-residual criterion within the established PaI framework. Broader architectural coverage can extend the study once the core mechanism has been fully characterized.
> > >
> > > Regarding practical motivation, PaI targets a goal that differs from the objective of designing compact architectures. Compact models improve inference cost through architectural redesign. PaI reduces training cost by constructing a sparse network at initialization that follows the same training recipe as the dense model. These two objectives contribute to efficiency in different ways. Compact architectures frequently rely on specialised training schemes or large-scale searches. PaI retains the training protocol while reducing parameter count before training begins. YOPO shows that a fully data-free low-rank-residual criterion can maintain accuracy comparable to existing PaI methods even on ImageNet and exceed them on smaller benchmarks. This demonstrates practical viability in scenarios where training data is limited, initialization cost must remain minimal, or transferable supermasks are required across tasks.
> > >
> > > ---
> > > [1] Hoang, D. N., Liu, S., Marculescu, R., and Wang, Z. Revisiting pruning at initialization through the lens of ramanujan graph. In International Conference on Learning Representations (ICLR’23), 2023.

---

### Author Response · Authors · 2025-12-02
**Rebuttal Summary for the Area Chair**

*We sincerely thank all the reviewers for their time, effort, and thoughtful assessment of our work.*

**Reviewer QzLy (Score: 8).**
The reviewer expressed no major concerns and indicated satisfaction with the clarifications provided.

> Reviewer's last comment: "Thank the authors for the rebuttal. I would be happy to maintain my score."

**Reviewer FeFf (Score: raised to 8).**
Following the inclusion of the additional comparison with NPB (Pham et al., NeurIPS 2023), the reviewer noted that YOPO achieves competitive accuracy while substantially improving computational efficiency, including reduced FLOPs and pruning speed improvements of 40 to 600 times relative to optimization-driven baselines. The reviewer also requested the computation of the dataset-transfer index (DTI, Definition 1) for SNIP and GraSP. The computed results show that data-dependent PaI methods, including SNIP and GraSP, produce high-risk masks with non-zero DTI for diverse datasets such as TinyImageNet, indicating weaker cross-dataset generalization. However, YOPO achieves near-zero DTI across all three benchmark datasets, producing "supermasks" even under extremely sparse conditions.

> Reviewer's last comment: "I thank the authors for conducting the new experiments on such short notice and providing additional clarifications. The method provides clear benefits either in performance or in terms of efficiency. I have improved my score to 8 based on the assumption that the experiments conducted during this phase will be extended to other networks / datasets and presented in the final paper.
"

**Reviewer tguL (Score: 2).**
The reviewer questioned the benchmark choices and suggested using ConvexNet. The rebuttal clarified that the benchmarks follow the current standards used in recent PaI publications (Xiang et al. (ICLR 2025), Ganjdanesh et al. (CVPR 2024), Cheng et al. (TPAMI 2024), Hoang et al. (NeurIPS 2023), and Pham et al. (NeurIPS 2023)). These works continue to rely on VGG, ResNet, WRN, CIFAR-10/100, TinyImageNet and ImageNet. ConvexNet does not appear in contemporary PaI literature, so it does not form a standard baseline for comparison.

> Discussion Phase Halt: "Due to detected leak of reviewer/AC identities"

**Reviewer mba5 (Score: 2).**
The reviewer initially interpreted YOPO as a post-training compression method (weakness 1). The rebuttal clarified that YOPO performs saliency estimation at initialization through low-rank residuals rather than compressing trained weights, and this clarification was accepted. The reviewer also remarked that YOPO does not surpass NPB in every setting. The response explained that YOPO and NPB differ at a foundational level: YOPO is a one-shot, data-free method based on structural decomposition, whereas NPB is an iterative, optimization-driven, and computationally intensive data-free method. YOPO reaches accuracy close to NPB while delivering pruning that is 40 to 600 times faster and retaining the lowest inference-time FLOPs. YOPO introduces a new design direction for zero-shot PaI, distinct from graph-based or optimization-based approaches, with headroom for further improvement through lightweight post-selection optimization. This positions YOPO as a strong foundation for future PaI developments.

> Discussion Phase Halt: "Due to detected leak of reviewer/AC identities"

---

**Overall contribution.**
YOPO introduces a zero-shot, data-free, and gradient-free pruning-at-initialization method grounded in nonnegative low-rank residuals, providing a principled and straightforward alternative to data-dependent initialization pruning. The method constructs a once-only saliency ordering that yields stable and reusable masks across sparsity budgets and datasets, supported by formal analysis of mask transferability through the Dataset Transfer Index (DTI Def. 1) and Initialization-Independence Index (I3, Def. 2). The nonnegativity-based residual structure supplies a lightweight collapse-avoidance mechanism that guarantees per-neuron survival without iterative scoring or layerwise optimization. Comprehensive experiments on CIFAR-10, CIFAR-100, Tiny-ImageNet, and ImageNet-1K using VGG, ResNet, and WRN architectures confirm that YOPO consistently outperforms existing zero-shot PaI approaches, achieves accuracy competitive with data-dependent baselines, and uniquely supports mask reuse without retraining the pruning criterion. YOPO reduces initialization cost by a factor of 40 to 600 compared to optimization-driven methods. It achieves the lowest inference-time FLOPs among the compared approaches, while maintaining competitive accuracy on large-scale benchmarks.

---

### Meta-Review · Area_Chair_5qdj · 2026-01-07

**Summary:**

This paper proposes a data/gradient-free pruning-at-initialization algorithm based on non-negative matrix factorization. Although the discussion period accidentally terminated, there have been extensive discussions between the authors and reviewers. Two negative reviewers did not change their scores after the discussions.

One reviewer was concerned that the baseline methods/network architectures in the experiments are outdated. While the authors argue that related works also use similar baselines, I agree with the reviewer's perspective. New and better methods are being developed year by year, and the baselines should change accordingly. If all researchers follow the previous baselines, then they will never change. Hence, I believe the reviewer's request to provide experiments with new network architectures is fair.

The other negative reviewer was not convinced about the motivation of the non-negative matrix factorization-based approach. While the theoretical motivation of a practical algorithm is typically described in a hand-wavy manner without thorough proofs, it seems that the authors failed to convince this reviewer during the discussion period.

Based on these reviews, I recommend the authors to resubmit this paper after adding more recent baselines and stating the theoretical motivation more clearly.

**Reviewer Concerns:**

Two major concerns are not solved:
- on the old baselines in experiments and
- unclear motivation of the proposed method.

There have been extensive discussions between authors and reviewers, but some reviewers are still concerning these issues.

**Reviewer Scores:**

Reviewer FeFf would increase their score from 4 to 8 as they promised in the discussion period. I do not expect other reviewers would change their scores.

---

### Decision · Program_Chairs · 2026-01-26

Reject